# Live-cell mapping of organelle-associated RNAs via proximity biotinylation combined with protein-RNA crosslinking

Pornchai Kaewsapsak[1,2,3,4†], David Michael Shechner[5,6,7†‡], William Mallard[5,6,7], John L Rinn[5,6,7§], Alice Y Ting[1,2,3,4,7]*

[1]Department of Chemistry, Massachusetts Institute of Technology, Cambridge, United States; [2]Department of Genetics, Stanford University, Stanford, United States; [3]Department of Biology, Stanford University, Stanford, United States; [4]Department of Chemistry, Stanford University, Stanford, United States; [5]Department of Stem Cell and Regenerative Biology, Harvard University, Cambridge, United States; [6]Department of Molecular and Cellular Biology, Harvard University, Cambridge, United States; [7]Broad Institute of Massachusetts Institute of Technology and Harvard, Cambridge, United States

*For correspondence:
ayting@stanford.edu

[†]These authors contributed equally to this work

Present address: [‡]Department of Pharmacology, The University of Washington, Washington, United States; [§]Department of Biochemistry, University of Colorado BioFrontiers, Colorado, United States

Competing interests: The authors declare that no competing interests exist.

**Abstract** The spatial organization of RNA within cells is a crucial factor influencing a wide range of biological functions throughout all kingdoms of life. However, a general understanding of RNA localization has been hindered by a lack of simple, high-throughput methods for mapping the transcriptomes of subcellular compartments. Here, we develop such a method, termed APEX-RIP, which combines peroxidase-catalyzed, spatially restricted in situ protein biotinylation with RNA-protein chemical crosslinking. We demonstrate that, using a single protocol, APEX-RIP can isolate RNAs from a variety of subcellular compartments, including the mitochondrial matrix, nucleus, cytosol, and endoplasmic reticulum (ER), with specificity and sensitivity that rival or exceed those of conventional approaches. We further identify candidate RNAs localized to mitochondria-ER junctions and nuclear lamina, two compartments that are recalcitrant to classical biochemical purification. Since APEX-RIP is simple, versatile, and does not require special instrumentation, we envision its broad application in a variety of biological contexts.
DOI: https://doi.org/10.7554/eLife.29224.001

## Introduction

Spatial compartmentalization of RNA is central to many biological processes across all kingdoms of life, and enables diverse regulatory schemes that exploit both coding and noncoding functions of the transcriptome. For example, the localization and spatially restricted translation of mRNA plays a fundamental role in a wide variety of biological contexts, including asymmetric cell division in bacteria and yeast, body-pattern formation in *Drosophila* and *Xenopus*, and signaling at mammalian neuronal synapses (*Jung et al., 2014*). Moreover, the localization of noncoding RNAs (ncRNAs) can play an architectural role in the assembly of subcellular structures, most notably within the nucleus, wherein ncRNAs help to assemble short-range chromatin loops, higher-order chromatin domains, and large sub-nuclear structures like nucleoli and Barr bodies, among others (*Rinn and Guttman, 2014*; *Engreitz et al., 2016*). However, despite these examples, our general understanding of the breadth and biological significance of RNA subcellular localization remains inchoate.

Techniques that elucidate the subcellular localization of RNAs are therefore critical for advancing our understanding of RNA biology. Classically, such techniques rely either on imaging or biochemical

fractionation. Imaging methods—such as Fluorescence In Situ Hybridization (FISH) and RNA reporter systems—are powerful tools for elucidating the positions of a small number of target RNAs at low-to-moderate throughput (*Wilk et al., 2016*; *Chen et al., 2015*; *Paige et al., 2011*; *Hocine et al., 2013*; *Nelles et al., 2016*; *Lécuyer et al., 2007*; *Garcia et al., 2007*). Alternatively, unbiased approaches for RNA discovery couple biochemical manipulations to microarray or deep sequencing analysis. For example, the RNA partners of proteins with characteristic subcellular localization can be identified through techniques that couple protein immunoprecipitation to RNA-Seq (*Ule et al., 2003*; *Gilbert et al., 2004*). Such methods have revealed the localization of many mRNAs, in addition to discovering novel non-coding RNAs involved in RNA splicing (*Chi et al., 2009*) and RNAi (*Motamedi et al., 2004*). On a broader scale, a deep sampling of RNAs residing within a cellular compartment—for example, an intact organelle of interest, or partitions along a sucrose gradient—can be identified by coupling subcellular fractionation to microarray analysis (*Diehn et al., 2000*, *2006*; *Marc et al., 2002*; *Sylvestre et al., 2003*; *Blower et al., 2007*; *Mili et al., 2008*; *Pyhtila et al., 2008*; *Chen et al., 2011*) or to RNA-Seq ('Fractionation-Seq,' *Sterne-Weiler et al., 2013*; *Mercer et al., 2011*). These powerful methodologies facilitate a deep characterization of the transcriptome of a subcellular target, in cases where a robust fractionation protocol for that target can be developed, and can sometimes be applied to native cells or tissues (*Diehn et al., 2006*).

Despite this progress, some technological gaps exist among current methods for studying RNA localization. Imaging approaches are of limited throughput, and may require specialized reagents, constructs, or microscopes that are only accessible to a handful of laboratories (*Wilk et al., 2016*; *Chen et al., 2015*; *Paige et al., 2011*; *Hocine et al., 2013*; *Nelles et al., 2016*). The efficacy of immunoprecipitation-based approaches is highly sensitive to the antibodies and enrichment protocols used (*Hendrickson et al., 2016*) and captures only RNAs that are directly complexed with each target protein. Fractionation-Seq is applicable only to organelles and subcellular fractions that can be purified, and—like all fractionation-based methods—can be complicated by contaminants and loss of material (*Lesnik and Arava, 2014*, *Lomakin et al., 2007*). Therefore, new technologies are needed for unbiased and large-scale discovery and characterization of RNA *neighborhoods*, with high spatial specificity, and within cellular structures that can be difficult to purify biochemically.

Here we introduce such a technology—termed APEX-RIP—that enables unbiased discovery of endogenous RNAs in specific cellular locales. APEX-RIP merges two existing technologies: APEX (engineered ascorbate peroxidase)-catalyzed proximity biotinylation of endogenous proteins (*Rhee et al., 2013*), and RNA Immunoprecipitation (RIP; *Gilbert et al., 2004*). We demonstrate that APEX-RIP is able to enrich endogenous RNAs in membrane-enclosed cellular organelles—such as the mitochondrion and nucleus—and in membrane-abutting cellular regions—such as the cytosolic face of the endoplasmic reticulum—although its applicability in completely unbounded compartments appears more limited. The specificity and sensitivity of this approach are higher than those obtained by competing methods. Moreover, by applying APEX-RIP to multiple mammalian organelles, we have generated high quality datasets of compartmentalized RNAs that should serve as valuable resources for testing and generating novel hypotheses pertinent to RNA biology. Given its ease of use and scalability across subcellular compartments, we anticipate that APEX-RIP will provide a powerful new tool for the study of RNA localization.

## Results

### Development of APEX-RIP and its application to mitochondria

APEX is an engineered peroxidase that can be targeted by genetic fusion to various subcellular regions of interest (*Rhee et al., 2013*) (*Figure 1A*). Upon addition of its substrates—biotin-phenol (BP) and hydrogen peroxide ($H_2O_2$)—to live cells, APEX catalyzes the formation of biotin-phenoxyl radicals that then diffuse outward and covalently biotinylate nearby endogenous proteins. More distal proteins are not significantly labeled because the biotin-phenoxyl radical has a half-life of less than one millisecond (*Wishart and Madhava Rao, 2010*). Previous work has shown that APEX-catalyzed proximity biotinylation, coupled to streptavidin enrichment and mass spectrometry, can generate proteomic maps of the mitochondrial matrix, intermembrane space, outer membrane, and nucleoid, each with <5 nm spatial specificity (*Rhee et al., 2013*; *Hung et al., 2014*, *2017*; *Han et al., 2017*).

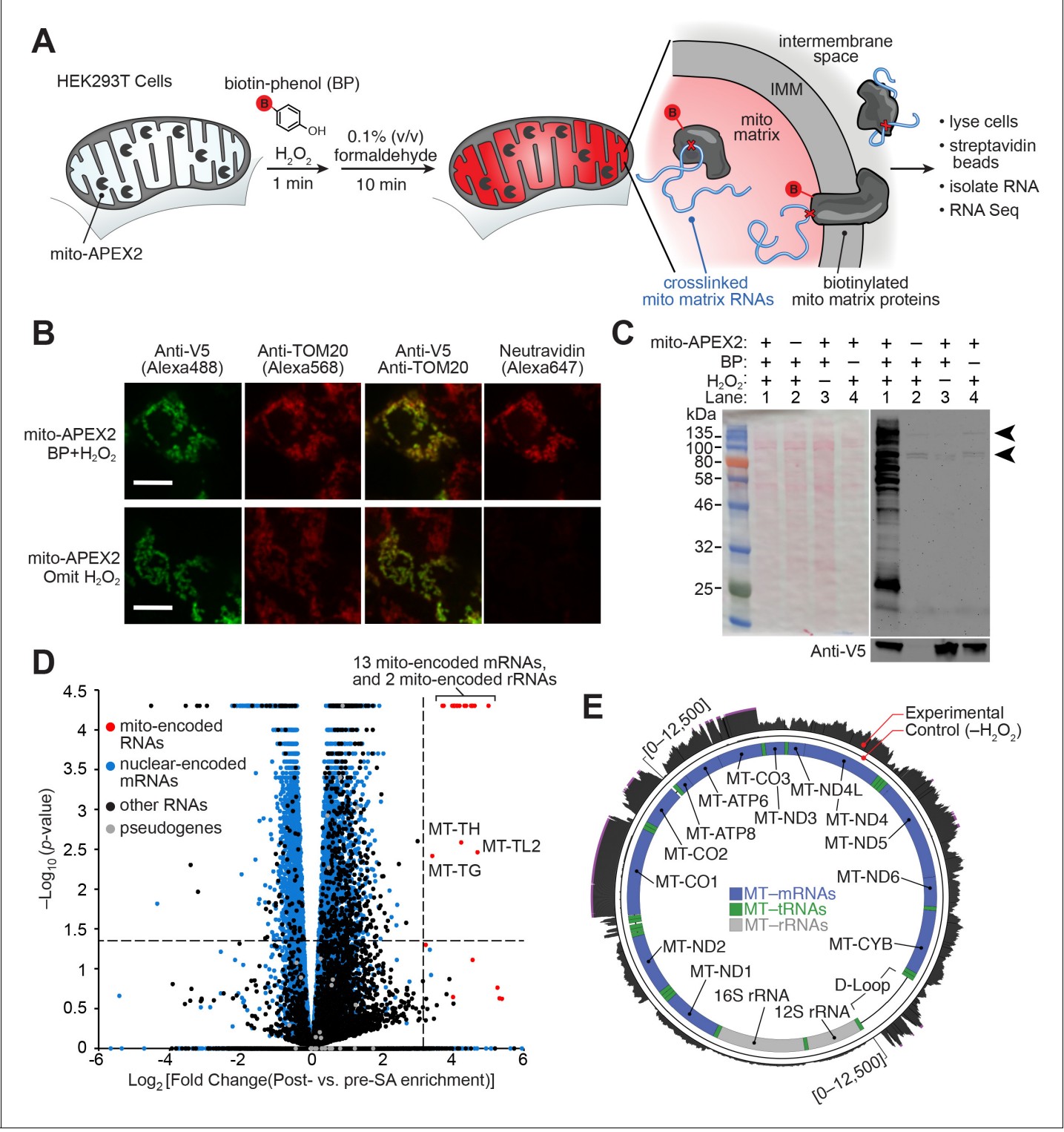

**Figure 1.** APEX-RIP in mitochondria. (**A**) Overview of the APEX-RIP workflow. Live cells expressing APEX2 (grey 'pacmen') targeted to the compartment of interest (here, the mitochondrial matrix) are incubated with the APEX substrate biotin-phenol (BP; red B: biotin). A one-minute pulse of $H_2O_2$ initiates biotinylation of proximal endogenous proteins (*Rhee et al., 2013*), which are subsequently crosslinked to nearby RNAs by 0.1% formaldehyde. Following cell lysis, biotinylated species are enriched by streptavidin pulldown, and coeluting RNAs are analyzed by qRT-PCR or RNA-Seq. IMM: inner mitochondrial membrane. (**B**) Imaging APEX2 biotinylation in situ. HEK 293T cells expressing V5-tagged mito-APEX2 were biotinylated using the APEX-RIP workflow, fixed, and stained as indicated. The bottom row is a negative control in which $H_2O_2$ treatment was omitted. Scale bars, 10 μm. TOM20 is a mitochondrial outer membrane protein; neutravidin staining detects biotinylation. (**C**) In situ biotinylation of the mitochondrial matrix proteome

*Figure 1 continued on next page*

*Figure 1 continued*

requires mito-APEX2, BP, and $H_2O_2$. Streptavidin blot analysis of whole cell lysates prepared following the protocol described in (**A**), or after omitting components of the APEX reaction. Arrowheads denote endogenous biotinylated proteins (*Chapman-Smith and Cronan, 1999*). Anti-V5 blot (*bottom*) detects expression of mito-APEX2. (**D–E**) mito-APEX-RIP efficiently recovers the mitochondrial transcriptome. (**D**) Gene-level RNA-Seq analysis of mito-APEX-RIP; data are the average values of three experimental replicates. Fold change is defined as ($FPKM_{post-enrichment}$/$FPKM_{pre-enrichment}$); dashed lines indicate significance thresholds for fold enrichment (determined by ROC analysis, see Materials and methods) and *p*-values calculated by CuffDiff2 (*Trapnell et al., 2013*). Mitochondrial genomes encode 13 mRNAs, two rRNAs and 22 tRNAs (*red*). Note that three mitochondrial tRNA genes, MT-TH, MT-TL2, and MT-TG, were also enriched. See *Supplementary file 1A*. (**E**) Nucleotide-level RNA-Seq analysis of mito-APEX-RIP, mapped to the human mitochondrial genome (innermost circle). Outermost circle: reads from the full APEX-RIP protocol; middle circle: reads from the negative control. Note the enrichment of several mitochondrially-encoded tRNAs and the D-loop leader transcript. Ribosomal RNAs were removed during library preparation (see Materials and methods). See also: *Figure 1—figure supplements 1,2*.

DOI: https://doi.org/10.7554/eLife.29224.002

The following figure supplements are available for figure 1:

**Figure supplement 1.** Optimization of APEX-RIP protocol.

DOI: https://doi.org/10.7554/eLife.29224.003

**Figure supplement 2.** Reproducibility of the mito-APEX2 RIP experiment.

DOI: https://doi.org/10.7554/eLife.29224.004

Because most cellular RNAs exist in close proximity to proteins, we reasoned that APEX-tagged subcellular proteomes could also provide access to the nearby subcellular transcriptomes by cross-linking labeled proteins and RNA together in situ (*Figure 1A*). As our first target organelle for this approach, we selected the mitochondrion because its RNA content—derived from both the mitochondrial genome and from imported, nuclear-encoded RNAs—has been extensively characterized by a wide array of complementary methods (*Mercer et al., 2011*; *Alán et al., 2010*; *Piechota et al., 2006*; *Ro et al., 2013*), hence providing a 'gold-standard' to which we can compare our results. The mitochondrial matrix was also the first mammalian compartment mapped by APEX proteomics methodology (*Rhee et al., 2013*). As an RNA-protein chemical crosslinker, we opted for mild formaldehyde treatment, which covalently captures most protein-protein and protein-nucleic acid interactions, and can be achieved with minimal disruption of native interactions in live cells. It is for these reasons that formaldehyde is used in several RIP technologies aimed at identifying the RNA partners of specific proteins of interest, including our own 'fRIP-Seq' protocol (*Chris and Svejstrup, 2006*, *Hendrickson et al., 2016*).

Since it was unclear *a priori* whether APEX-catalyzed biotinylation should precede or follow the formaldehyde crosslinking step, we explored both schemes in parallel (*Figure 1—figure supplement 1A*; see Materials and methods). Each protocol, applied to HEK 293T cells that transiently expressed mitochondrially-localized APEX ('mito-APEX,' *Supplementary file 5A*), resulted in clear enrichment of fifteen mitochondrial-encoded RNAs—relative to the cytosolic marker *GAPDH*—as gauged by qRT–PCR (average of 49.3 ± 3.5 and 60.9 ± 4.1 fold enrichment, respectively, *Figure 1—figure supplement 1A*). We next proceeded to RNA-Seq analysis, assuming that fixing cells prior to biotinylation would better capture transient or weak RNA–protein interactions, and therefore selecting the crosslinking-then-BP protocol (see Materials and methods). However, since it was unknown whether biotin-phenoxyl radicals might cleave or modify RNA in a manner that introduces bias into deep-sequencing libraries (*Ziehler and Engelke, 2000*), we chose to prepare these libraries using the 'Ribo-Zero' method, which physically removes ribosomal RNAs prior to fragmentation and sequencing adaptor ligation (Materials and methods). Since this workflow does not require the presence of a 3′–poly(A) tail for first-strand synthesis, it offers superior coverage in cases with lower input quality (*Adiconis et al., 2013*), and furthermore enables sampling of a broader range of RNA classes.

Deep-sequencing of mito-APEX-RIP libraries confirmed that mitochondrial mRNAs were substantially enriched over the majority of nuclear-encoded genes. However, a sizeable 'shoulder'—comprising a number of conspicuous off-target RNAs—was also unexpectedly observed (*Figure 1—figure supplement 1B,C*). To address this issue, we re-examined our labeling and crosslinking protocols, using a sampling of these off-target RNA markers (e.g., the abundant nuclear RNA *XIST*, and cytosol-localized RNAs *HOOK2* and *MAN2C1*) as more incisive negative controls. We furthermore employed HEK293T cells that stably expressed mitochondrially-localized APEX2 (mito-APEX2,

*Figure 1B–C*, *Supplementary file 5A-B*), a more active APEX variant that we hypothesized might improve target enrichment (*Lam et al., 2015*). This improved construct, and more controlled analysis revealed that APEX labeling followed by crosslinking provides superior specificity, improving the average enrichment of target RNAs—relative to the contaminant RNAs identified above—by nearly ten-fold (*Figure 1—figure supplement 1C*). We suspect that the mild formaldehyde treatment compromises membrane integrity (*Fox et al., 1985*), allowing BP radicals to escape to adjoining compartments when APEX labeling is performed after, rather than before, formaldehyde treatment.

Using the optimized APEX-first/crosslinking-second protocol, we then mapped the mitochondrial transcriptome of mito-APEX2-expressing HEK 293T cells by RNA-Seq (*Figure 1D*, *Supplementary file 1B*). Gene-level analysis comparing fold enrichment and statistical significance of all human genes (Materials and methods) revealed that all 13 mRNAs and both rRNAs encoded by the mitochondrial genome were highly enriched (greater than 11-fold; *Figure 1D* and *Figure 1—figure supplement 2*, *Supplementary file 1A*). Surprisingly, we even observed the enrichment of several mitochondrial-encoded tRNAs, although our library preparation workflow generally excluded such smaller RNA species (*Figure 1D*). Read density plots mapped to the mitochondrial genome demonstrated that most of our captured RNAs correspond to fully-processed transcripts, including mRNAs, interstitial tRNAs, and the D-loop leader sequence from which mitochondrial transcription initiates (*Figure 1E*). Intriguingly, mito-mRNA read densities appeared to correlate with previous measures of mRNA half-life (*Nagao et al., 2008*). For example, mRNAs encoding MTCO1-3 have longer half-lives, and more reads from APEX-RIP, than mRNAs encoding MTND1-2. We therefore conclude that APEX-RIP is a specific and sensitive approach for mapping the transcriptome within a membrane-bound organelle.

## APEX-RIP mapping of nuclear-cytoplasmic RNA distribution

Having established that APEX-RIP in the mitochondrion, we next turned our attention to a more challenging compartment: the mammalian nucleus. The nucleus is more complex and has a less well-defined transcriptome than the mitochondrial matrix, but previous Fractionation-Seq datasets from HEK 293T (*Sultan et al., 2014*) again provide a reference list to which we can compare our results.

We generated HEK 293T cells that stably express APEX2 in the nucleus (APEX-NLS) or in the cytosol (APEX-NES, where NES is a Nuclear Export Signal) (*Supplementary file 5A*). The specificity of in situ biotinylation by these constructs within each compartment was confirmed by imaging (*Figure 2A*, *Supplementary file 5B*). Whole cell lysates prepared from each cell line also produced distinct 'fingerprints' of biotinylated proteins, as assayed by streptavidin blotting (*Figure 2—figure supplement 1*).

We performed APEX-RIP on both APEX-NLS and APEX-NES cells, using the biotinylation-first/crosslinking-second protocol established above, with an additional one-minute radical-quenching step in between the APEX and crosslinking steps (*Figure 2—figure supplement 2*; see Materials and methods). Encouragingly, 'gold standard' nuclear and cytosolic RNAs were enriched from the corresponding cell lines as predicted: long non-coding RNAs, which are predominantly nuclear, were enriched in APEX-NLS-RIP and de-enriched in APEX-NES-RIP (*Figure 2B*, *top*), while endoplasmic reticulum-proximal mRNAs (*Jan et al., 2014*) exhibited the converse profile (*Figure 2B middle*). As a further test, we directly compared the enrichments from APEX2-NLS and APEX2-NES to one another, confirming that they had effectively parsed known nuclear- and cytosol-localized RNAs into the expected compartments (*Figure 2B bottom* and C). We used Receiver Operating Characteristic (ROC) analysis to obtain final transcript lists of 5740 nuclear RNAs and 5367 cytosolic RNAs, with observed contamination frequencies (i.e. the ratio of enriched off-target RNAs to total enriched RNAs) of <1.6% and <1.5%, respectively (*Supplementary file 2A-C*, *Figure 2—figure supplement 3A–B*, see Materials and methods).

Surprisingly, we also observed sizeable populations of RNAs exhibiting noncanonical nuclear–cytoplasmic partitioning. 3161 mRNAs—including *C1orf63*, for example (*Figure 2D*, *top right*)—appeared preferentially nuclear. Many of these species have been proposed to play a role in dampening gene expression noise (*Bahar Halpern et al., 2015*). Conversely, 81 lncRNAs appeared preferentially cytoplasmic (*Figure 2D*, *bottom left*); these include the known cytoplasmic lncRNA *SNHG5*, a modulator of staufen-mediated decay that influences colorectal tumor growth (*Derrien et al., 2012*; *Damas et al., 2016*) (*Figure 2D*, *bottom right*). We were concerned that this atypical RNA localization might be artifactual, since diffusion of proteins between subcellular compartments

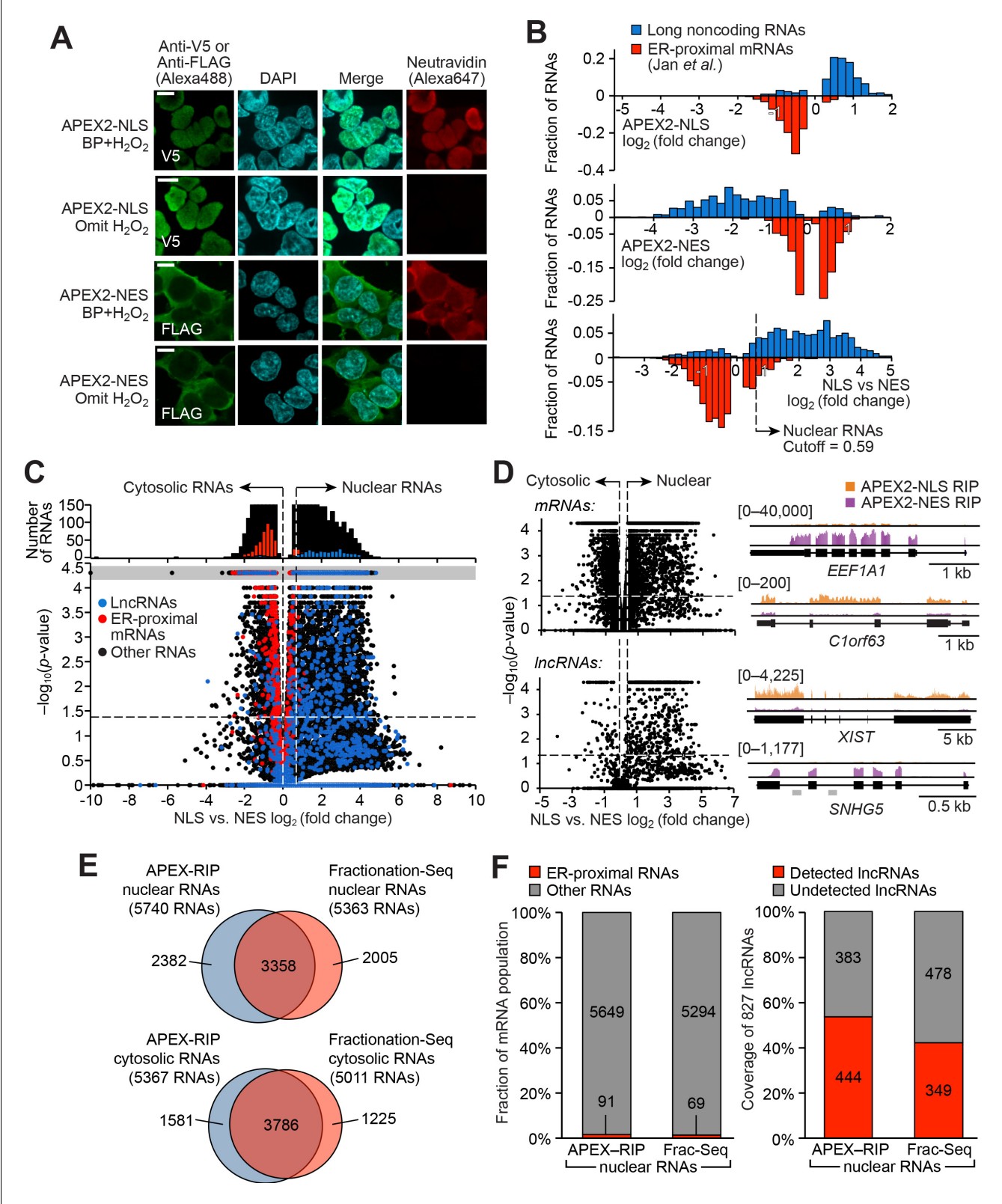

**Figure 2.** APEX-RIP mapping of the nuclear-cytoplasmic RNA distribution. (**A**) Fluorescence imaging of nuclear and cytosol-targeted APEX2 fusion constructs. HEK 293T cells expressing the indicated constructs ('NLS,' nuclear localization signal; 'NES,' nuclear export signal) were labeled with biotin-phenol, crosslinked and stained as indicated. Scale bars, 10 μm. DAPI is nuclear stain. (**B**) APEX-RIP recovers known nuclear and cytosolic standard RNAs, defined here as long noncoding RNAs (nuclear markers, *blue*) and RNAs proximal to the Endoplasmic Reticulum (cytoplasmic markers, *red*—

*Figure 2 continued*

defined by (*Jan et al., 2014*), with measured $p$-values$\leq$0.05—see Materials and methods). *Top*: APEX2–NLS-RIP enriches nuclear standards. *Middle*: APEX2–NES-RIP enriches cytoplasmic standards. *Bottom*: Combined analysis of the APEX2–NLS and APEX2–NES RIP experiments distinguish the two classes. Fold changes are defined as (FPKM$_{post-enrichment}$/FPKM$_{pre-enrichment}$); combined fold change as [FPKM$_{NLS-post-enrichment}$/FPKM$_{NES-post-enrichment}$]. Dotted line indicates the significance threshold for nuclear localization. (C) Global analysis of nuclear and cytoplasmic RNA localization by combined APEX2-NLS and APEX2-NES RIP. Vertical dashed lines indicate the cutoffs for nuclear and cytosolic RNAs. Horizontal dash line indicates $p$-value=0.05. Top histogram illustrates the distribution of RNAs with $p$-value=$5\times10^{-5}$, which are boxed in gray in the scatter plot. The average of data from three biological replicates are shown. See *Supplementary file 2A-B*. (D) APEX-RIP reveals classes of RNAs with canonical and noncanonical nuclear-cytoplasmic distributions. Left: the same data as in (C), separately parsed into mRNAs (*top*) and lncRNAs (*bottom*). Right: read density plots of example RNAs from each class that exhibit stereotypical and atypical localization. *EEF1A1* and *C1orf63* are mRNAs; *XIST* and *SNHG5* are lncRNAs. For each gene, a common y-scale is used for all read tracks. SnoRNAs encoded in the *SNHG5* gene body are indicated as gray rectangles. (E) Venn diagrams comparing APEX-RIP and fractionation-based RNA datasets (*Sultan et al., 2014*). (F) Nuclear APEX-RIP is more sensitive than is biochemical fractionation. *Left:* Specificity of the APEX-RIP and nuclear RNA datasets (*Sultan et al., 2014*). Off-target RNAs were defined as actively translated ER-proximal mRNAs (*Jan et al., 2014*). *Right:* Recall of nuclear standard RNAs, defined as a set of 827 lncRNAs annotated by GENCODE hg19 with average pre-enrichment FPKM $\geq$ 1.0. See also: *Figure 2—figure supplements 1–2*.

DOI: https://doi.org/10.7554/eLife.29224.005

The following figure supplements are available for figure 2:

**Figure supplement 1.** Characterization of APEX2 fusion constructs.
DOI: https://doi.org/10.7554/eLife.29224.006
**Figure supplement 2.** Reproducibility of nuclear–cytoplasmic APEX-RIP experiments.
DOI: https://doi.org/10.7554/eLife.29224.007
**Figure supplement 3.** Precision and specificity of nuclear–cytoplasmic APEX-RIP, and its comparison to subcellular fractionation.
DOI: https://doi.org/10.7554/eLife.29224.008

during a ten-minute formaldehyde treatment might allow aberrant RNA-protein interactions to be chemically crosslinked. To rule out this possibility, we monitored the localization of APEX-labeled species during the course of a BP-first/crosslink-second NLS-APEX2-RIP experiment, and failed to observe significant migration of biotinylated proteins from the nucleus into the cytosol (*Figure 2—figure supplement 3C*).

Our APEX-RIP nuclear and cytosolic RNA lists provide an opportunity for a head-to-head comparison with the traditional Fractionation-Seq method for mapping subcellular RNA localization. ROC analysis of HEK 293T fractionation-Seq data obtained using library synthesis and sequencing protocols very similar to our own (see Materials and methods, *Sultan et al., 2014*) yielded 5363 nuclear RNAs and 5011 cytosolic RNAs enriched by fractionation (*Figure 2—figure supplement 3D–G*; *Supplementary file 2D-F*). Of these RNAs, 63% (3358) were also enriched in our APEX-RIP nuclear dataset, implying general agreement between the two technologies (*Figure 2E*). Notably, APEX-RIP also enriched nearly 2400 additional transcripts. These may be nuclear-localized RNAs that were opaque to the fractionation protocol, or contaminants enriched by APEX-RIP. To address this latter possibility, we examined each dataset for conspicuous non-nuclear contaminants: RNAs that are known to be localized at the Endoplasmic Reticulum (*Jan et al., 2014*). Satisfyingly, each nuclear dataset exhibited similarly low levels of ER contaminants (1.6% and 1.3%, respectively, *Figure 2F*, *left*).

To compare the coverage, or sensitivity, of each method (sometimes termed recall), we examined the enrichment in each dataset of lncRNAs, which are thought to be predominantly nuclear (*Derrien et al., 2012*). We assembled a list of 827 annotated lncRNAs (GENCODE v19) with average pre-enrichment FPKM greater than 1.0 (*Supplementary file 2G*). Of these lncRNAs, 53.6% are enriched in our APEX-RIP-derived nuclear dataset, while nuclear Fractionation-Seq from the same cell line enriched only 42.2% (*Figure 2F*, *right*). We conclude that APEX-RIP rivals or outperforms Fractionation-Seq in terms of both specificity and coverage, for analysis of endogenous RNA subcellular localization.

## Enrichment of RNAs proximal to the ER membrane

Having established that APEX-RIP can enrich RNAs in membrane-enclosed cellular compartments, we next sought to address whether the technique could successfully capture the transcriptomes of 'open' subcellular regions. Previous proteomic work has shown that APEX tagging exhibits sufficient spatial specificity for such open compartments, since this technology has produced highly specific

proteomic maps of, for example, the mammalian neuronal synaptic cleft (*Loh et al., 2016*), outer mitochondrial membrane (*Hung et al., 2017*), mitochondrial nucleoid (*Han et al., 2017*), and G-protein coupled receptor interaction networks (*Lobingier et al., 2017*; *Paek et al., 2017*). We were unsure, however, if the additional formaldehyde crosslinking step would preserve or blur the estimated <5 nanometer spatial resolution of APEX labeling.

As a test case for the generality of APEX-RIP at such open compartments, we selected the Endoplasmic Reticulum (ER). The ER is an appealing target for several reasons. First, it is host to a known set of characteristic RNAs that we can use as positive controls—the so-called 'secretome'—which comprises mRNAs encoding secreted, glycosylated, and/or transmembrane proteins that are translated on the rough ER. Second, the ER provides the opportunity to compare the efficacy of APEX-RIP to alternative approaches, since RNAs in this subcellular locale have been previously characterized both by Fractionation-Seq, and by a newer methodology termed proximity-dependent ribosome profiling (*Jan et al., 2014*; *Williams et al., 2014*). This latter technique maps active protein translation at the ER membrane by combining ribosome profiling (*Ingolia et al., 2009*) with proximity-restricted sequence-specific biotinylation, using an ER-targeted biotin ligase and ribosomes that are tagged with the peptide substrate (AviTag) of that ligase. Although the library preparation protocols used in each of these studies varied significantly from our own (see Materials and methods), by focusing our analyses on the fold enrichment of transcripts between matched input and ER-bound samples—and not on absolute transcript abundances—we hoped to control for these differences.

Since it was initially unclear which face of the ER membrane (cytosolic or luminal) would be most amenable to the APEX-RIP method, we generated fusion constructs that localized the peroxidase catalytic center to each (*Figure 3A–B*, *Supplementary file 5A*). ERM-APEX2 targets APEX2 to the ER cytosolic surface via a 27-amino acid fragment derived from the native ER membrane (ERM) protein cytochrome P450 C1. HRP-KDEL targets horseradish peroxidase (HRP) to the ER lumen via an N-terminal ER-targeting signal and a C-terminal KDEL ER-retention motif (*Martell et al., 2012*). We have shown that HRP catalyzes the same proximity-dependent biotinylation chemistry as APEX2 (*Loh et al., 2016*), but has higher specific activity than APEX2 in the ER lumen (*Lam et al., 2015*). We generated HEK 293T cells stably expressing ERM-APEX2 and HRP-KDEL, and confirmed by microscopy and streptavidin blotting that each produced the expected labeling patterns (*Figure 3C and D*, *Figure 2—figure supplement 1*; *Supplementary file 5B*. *see also Hung et al., 2017*), *Figure 1D*). Next, we compared the efficacy of each construct for target RNA isolation, using the biotinylation-first/crosslinking-second APEX-RIP protocol, and analyzing our results via qRT-PCR analysis of established secretome and non-secretome mRNAs (*Jan et al., 2014*). Parallel experiments with APEX2-NES cells served as negative controls (*Figure 3E*, *Supplementary file 5C*).

Intriguingly, while APEX-RIP from HRP-KDEL cells efficiently enriched target secretome mRNAs relative to non-target controls (average fold enrichment = 19.5, two-tailed t-test $p$-value = 0.00009), parallel experiments in ERM-APEX2 cells exhibited only modest, qualitative enrichment of target species (average fold enrichment = 1.49, two-tailed t-test $p$-value = 0.0515). Indeed, results from ERM-APEX2 cells were nearly indistinguishable from those acquired from APEX2-NES control cells (Student's two-tailed t-test comparing the two constructs $p$-value = 0.830, *Figure 3E*, *right*). This is surprising since proteomic experiments in HEK 293T cells expressing the identical ERM-APEX2 construct yielded highly specific enrichment of ER-localized proteins (*Hung et al., 2017*).

Our data strongly imply that APEX-RIP does not have the same spatial specificity as peroxidase-catalyzed proteomic labeling, and may be limited by perturbations induced by formaldehyde crosslinking. However, we were highly encouraged by the data obtained with the HRP-KDEL construct, which we ascribe to the lower diffusion rates of both proteins and biotin-phenoxyl radicals when constrained within the limits of the ER lumen. We thus hypothesize that APEX-RIP with this construct is effective because formaldehyde crosslinking physically couples RNAs on the cytosolic face of the ER to protein complexes that are biotinylated within the ER lumen, thereby allowing target RNAs to be enriched by streptavidin (*Figure 3A*). Furthermore, we observed that the target specificity of this approach could be greatly improved by addition of a one-minute radical-quenching step in between the biotinylation and crosslinking steps in our protocol (*Figure 3—figure supplement 1A*). We surmise that this additional step prevents residual peroxidase-generated radicals from leaking into adjoining compartments when the integrity of the ER membrane is compromised during formaldehyde treatment.

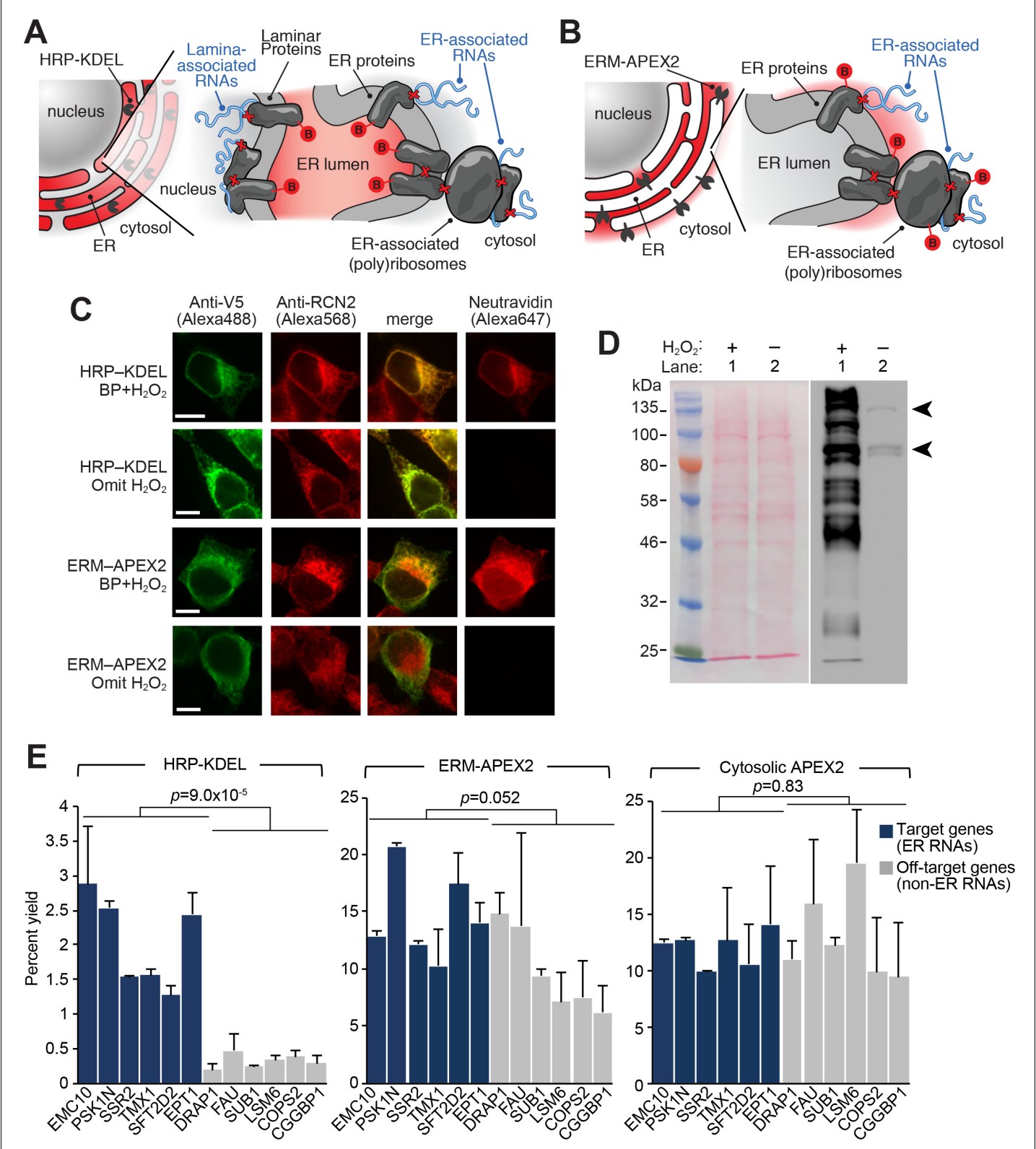

**Figure 3.** APEX-RIP at the Endoplasmic Reticulum membrane. (**A—B**) Schematics summarizing alternating ER-targeting strategies. (**A**) HRP, targeted to the ER lumen with a KDEL sequence, biotinylates proteins within the ER. Red B: biotin. Red X's: chemical crosslinks induced by 0.1% formaldehyde treatment. Note that RNAs enriched by this approach may reside at the cytosolic face of the ER, or at the nuclear lamina, as shown. (**B**) APEX2, targeted to the ER membrane (ERM) by fusing it to the transmembrane segment of rabbit P450 C1, biotinylates proteins proximal to the cytosolic face of the ER.

*Figure 3 continued on next page*

*Figure 3 continued*

(**C**) Imaging of biotinylation from HRP-KDEL and ERM-APEX2 catalyzed reactions. HEK293T cells stably expressing HRP-KDEL or ERM-APEX2 were labeled with BP, fixed and imaged as in *Figure 1B*. Scale bars, 10 μm. Anti-RCN2 was used to mark ER lumen. (**D**) Streptavidin blot detection of resident ER proteins biotinylated by HRP-KDEL, as in *Figure 1C*. Arrowheads denote endogenously biotinylated proteins (*Chapman-Smith and Cronan, 1999*). (**E**) qRT-PCR analysis, comparing the specificities of the labeling schemes shown in (**A–B**). Target and off-target genes were selected using previously-reported RNA abundances at the ER membrane (*Jan et al., 2014*). Cytosolic APEX2 (APEX2–NES, as in *Figure 2*) serves as a negative control. Data are the mean of four replicates ± one standard deviation. Significance testing: Student's two-tailed t-test.

DOI: https://doi.org/10.7554/eLife.29224.009

The following figure supplements are available for figure 3:

**Figure supplement 1.** Further optimization of the APEX-RIP protocol; additional HRP-KDEL RIP data.
DOI: https://doi.org/10.7554/eLife.29224.010
**Figure supplement 2.** Additional analysis comparing HRP-KDEL RIP data and other ER-RNA data sets.
DOI: https://doi.org/10.7554/eLife.29224.011

Using this improved protocol, we performed APEX-RIP on HRP-KDEL cells (*Figure 3—figure supplement 1B–C*, *Supplementary file 3B*). Gene-level analysis, comparing RNA counts before and after streptavidin pulldown, revealed that the majority (72%) of secretome mRNAs expressed in our cells (defined by ER-proximal RNAs (*Jan et al., 2014*) and Phobius-predicted mRNAs with exclusion of nuclear-encoded mitochondrial mRNAs, see Materials and methods) were enriched, while mRNAs in a test set of known non-secreted genes were not enriched, thus demonstrating the ability of our method to isolate ER-associated transcripts from the larger population of cellular RNAs (*Figure 4A*). Using p-values and ROC analysis, we determined the optimal $\log_2$ fold change significance threshold (*Figure 3—figure supplement 1D–F*; see Materials and methods), obtaining a final list of 2672 ER-associated RNAs that were significantly enriched in multiple experiments (*Figure 4B*; *Supplementary file 3A*). We did not detect any obvious trend among the 28% of expressed secretome mRNAs that were not represented in this list. However, this dataset exhibited 96.5% specificity, based on previous secretory annotation as defined by GOCC, SignalP, TMHMM, or Phobius (*Ashburner et al., 2000*; *Petersen et al., 2011*; *Krogh et al., 2001*; *Käll et al., 2004*), while mRNAs lacking such signals were concomitantly depleted (*Figure 4C*). Coverage was likewise exceptional (97%), as gauged by the successful recall of 71 mRNAs encoding well-established ER resident proteins (*Figure 4D*, *Supplementary file 3E*; see Materials and methods).

We next compared the KDEL-APEX-RIP ER-associated RNA dataset to analogous results obtained by subcellular biochemical fractionation (*Reid and Nicchitta, 2012*), and by proximity-dependent ribosome profiling (*Jan et al., 2014*) (*Supplementary file 3C-D*, respectively). Encouragingly, KDEL-APEX-RIP captures the majority of RNAs enriched by each of these alternative techniques (69% and 97%, respectively, *Figure 4E*), implying broad agreement between the different methodologies. To examine this further, we quantified the specificity and coverage of each approach, as above (see Materials and methods). Specificity analysis demonstrated that APEX-RIP and ribosome profiling exhibited similarly high specificity (96.5% and 99.2%, respectively). However, Fractionation-Seq was substantially noisier, such that only 91% of enriched mRNAs bore a secretory annotation (*Figure 4C*); the remaining 9% comprised sizeable populations of conspicuous contaminants (*Figure 3—figure supplement 2A*). The coverage of ER-localized mRNAs retrieved by APEX-RIP (97%) was also considerably higher than those retrieved by both Fractionation-Seq and ribosome profiling (73% and 70%, respectively, *Figure 4D*). We attribute the enhanced coverage of APEX-RIP to its higher sensitivity, since this method appears better suited for capturing RNAs with lower abundances than the alternative approaches. Of the transcripts enriched by Frac-Seq or ribosome profiling, 95% have input abundances of 3.68 and 6.49 FPKM or higher, respectively, whereas those enriched by APEX-RIP have an analogous lower expression limit of 0.42 FPKM (*Figure 3—figure supplement 2B*). Such higher sensitivity may also explain why the set of RNAs enriched by APEX-RIP is so much larger than those obtained by fractionation and ribosome-profiling (*Figure 4E*). Excitingly, this further underscores the ability of APEX-RIP to recover RNAs that are opaque to other methods. While the vast majority (93.3%) of our enriched RNAs are mRNAs, we also enrich dozens of noncoding RNA species—including antisense RNAs and lincRNAs (*Figure 4B*). These RNAs are not translated, and thus cannot be detected by ribosome profiling, and tend to be lowly expressed, making them difficult targets for either ribosome profiling or Fractionation-Seq (*Figure 3—figure supplement*

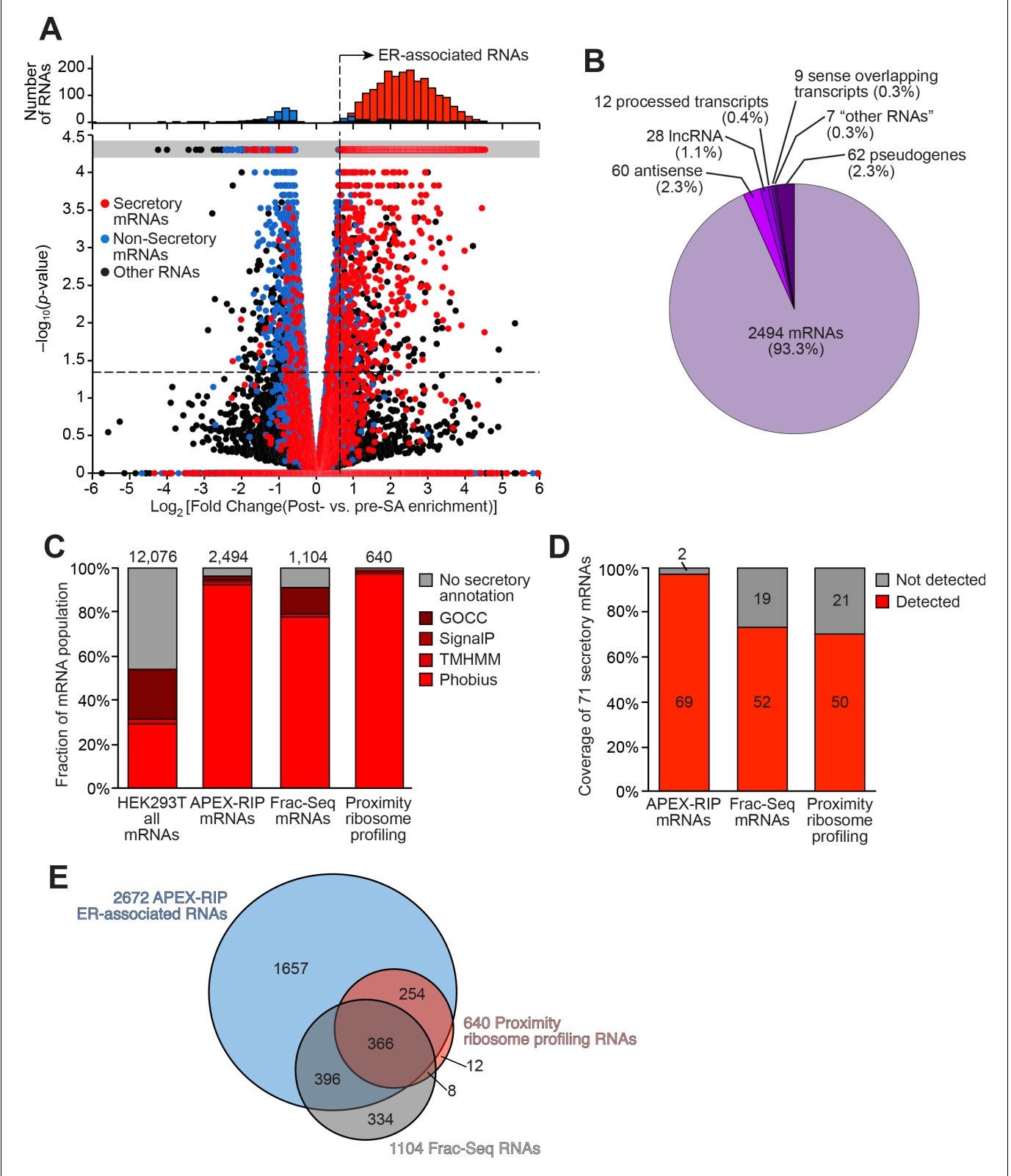

**Figure 4.** Mapping the ER-proximal transcriptome with APEX–RIP. (**A**) Global analysis of RNA localization at the Endoplasmic Reticulum. Fold change = (FPKM$_{post-enrichment}$/FPKM$_{pre-enrichment}$). Horizontal dashed line indicates p-value = 0.05. Top histogram illustrates the distribution of RNAs with p-value = $5 \times 10^{-5}$, which are boxed in gray in the scatter plot. Average data from two biological replicates are shown. Standard mRNAs encoding known secretory and non-secretory proteins are highlighted in red and blue, respectively (see Materials and methods). (**B**) Classification of APEX-RIP

*Figure 4 continued on next page*

*Figure 4 continued*

enriched, ER-associated RNAs. Collectively, all classes of non-coding RNAs constitute 6.7% of enriched genes (178 of 2672 RNAs). (C) Specificity analysis for protein-coding mRNAs in the APEX-RIP-derived ER-associated RNA list. The total number of RNA species in each condition is indicated above each column. 96.5% of the 2494 APEX-RIP ER-enriched mRNAs exhibit some form of secretory annotation (as predicted by Phobius, TMHMM, SignalP, or GOCC, *see methods*), whereas only 53.8% of total mRNAs expressed in HEK293T cells (FPKM $\geq$ 1.0) are similarly classified (*left*). (D) Target recall of ER APEX-RIP exceeds those of proximity-restricted ribosome profiling (*Jan et al., 2014*: see *Supplementary file 3D*) and biochemical fractionation (*Reid and Nicchitta, 2012*; see *Supplementary file 3C*). See also: *Supplementary file 3E*. (E) Venn diagram comparing RNA datasets. Note that all enriched RNAs in Reid et al. ER fractionation-Seq dataset were mRNAs. See also: *Figure 3—figure supplements 1–2*.

DOI: https://doi.org/10.7554/eLife.29224.012

*2B*). While some proportion of these hits may constitute experimental noise, we believe the remainder hint at unanticipated roles for noncoding RNAs at the ER.

In summary, APEX-RIP is a powerful method for mapping endogenous RNAs proximal to the ER membrane, with a sensitivity and precision that equals or surpasses alternate technologies. We anticipate that this approach may be extensible to other membrane-abutting subcellular regions as well.

## Hypotheses from ER and nuclear APEX-RIP datasets

We wondered if the RNA subcellular localization datasets produced by APEX-RIP could be mined for new biological hypotheses. To explore this possibility, we sought to computationally identify potential candidate RNAs that are localized at the interfaces between cellular compartments, since such transcripts are difficult to isolate by conventional approaches. We focused on two such interfaces: the ER-mitochondrial junction and the nuclear lamina.

We sought to identify RNAs localized to the ER-mitochondrial junction through close inspection of our KDEL dataset. It is thought that that the bulk of the nuclear-encoded mitochondrial proteome is translated either within the cytosol, or in proximity to mitochondria themselves (*Lesnik et al., 2015*). However, of the 2494 mRNAs in our ER-associated RNA dataset, 135 code for mitochondrial proteins, as defined by GOCC. Since the majority of these genes (132 mRNAs, 98%) also carry secretory annotation, we considered the possibility that the translation or processing of these 135 mRNAs require machinery localized at the ER membrane. For example, these mRNAs might be translated at mitochondria-ER contact sites, some of which have been observed to contain ribosomes (*Csordás et al., 2006*). To gain initial insight into these unusual RNAs, we analyzed these genes to see whether, relative to total pool of mRNAs encoding mitochondrially-localized proteins, they were enriched in particular characteristics (*Supplementary file 4A*). Intriguingly, 62.7% of these mRNAs code for transmembrane proteins (as predicted by TMHMM), compared to only 20.4% of all nuclear-encoded mitochondrial genes (*Figure 5A*). Subcompartment analysis of this ER-proximal population was also revealing: of the 39 genes for which compartment-specific annotations were available, 49% (19 genes) encode proteins destined for the outer mitochondrial membrane (OMM), whereas OMM proteins comprise only 18% of the bulk mitochondrial proteome (*Figure 5B*). This may indicate something unique about the biogenesis of OMM proteins, since the mRNAs encoding IMM-destined proteins did not exhibit such enrichment (comprising ~41–44% of both our ER-proximal population, and the general mitochondrial proteome), and those encoding matrix and intermembrane space proteins were depleted in our set (*Figure 5B*). Interestingly, in yeast, proximity-dependent ribosome profiling near the OMM showed similar enrichment of mRNAs encoding proteins destined for the inner mitochondrial membrane (*Williams et al., 2014*). Perhaps a subset of proteins destined for both the outer and inner mitochondrial membranes are locally translated at mitochondria-ER contact sites.

We adopted a slightly different computational approach to identify candidate nuclear laminar RNAs—transcripts that have long been proposed to contribute to the laminar functions of gene repression (*Kind et al., 2010*) and nuclear architecture (*Chen et al., 2016*), but for which few examples have been identified. Because intermembrane space of the nuclear envelope is contiguous with the ER lumen, we hypothesized that our KDEL-APEX-RIP experiment—in addition to enriching RNAs proximal to the ER—might also enrich RNAs at the nuclear lamina (*Figure 3A*). We therefore sought to discover candidate laminar RNAs by examining the population of KDEL-enriched RNAs for transcripts that are predominantly nuclear—that is, by intersecting our ER-associated and nuclear RNA lists (*Figure 5C*). When we performed this analysis, and filtered this intersected list to remove

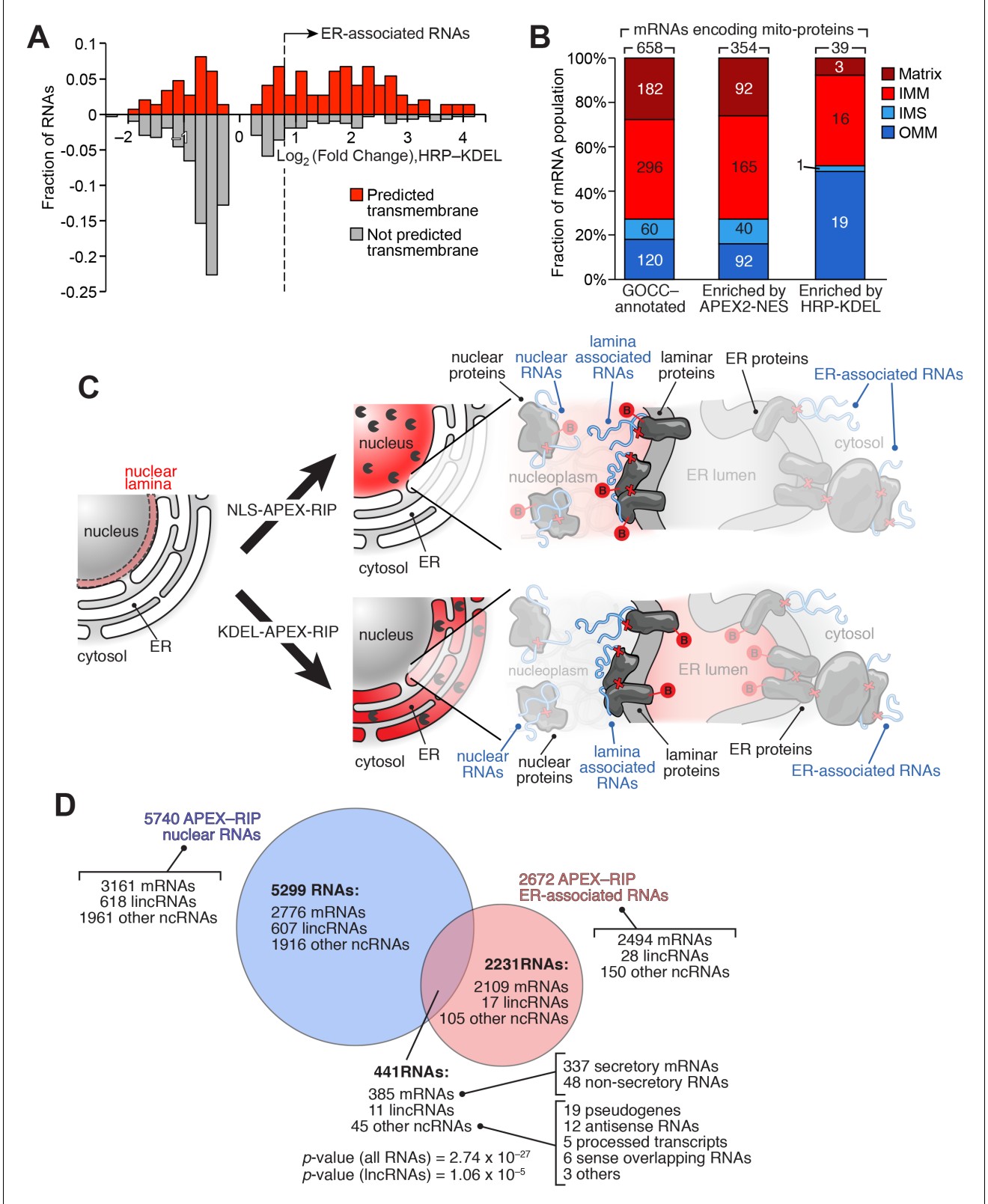

**Figure 5.** APEX-RIP reveals RNAs with potentially novel localization. (**A**) Many mitochondrial transmembrane proteins appear to be translated at the ER. mRNAs encoding mitochondrial proteins (defined by GOCC and MitoCarta 1.0 (*Ashburner et al., 2000*; *Pagliarini et al., 2008*) with predicted transmembrane helices (predicted by TMHMM (*Krogh et al., 2001*); red distribution) are preferentially enriched by HRP-KDEL APEX-RIP, relative to those encoding mitochondrial proteins lacking transmembrane domains (gray distribution). See *Supplementary file 3A*. (**B**) Mitochondrial proteins
*Figure 5 continued on next page*

*Figure 5 continued*

encoded by ER-proximal mRNAs are enriched for outer mitochondrial membrane (OMM) destined proteins, and de-enriched for matrix and intermembrane space (IMS) destined proteins. Predicted sub-mitochondrial localization of all GOCC-annotated mitochondrial proteins (*left*), those with mRNAs enriched by APEX2–NES (*middle*), and those enriched by HRP–KDEL (*right*). IMM: Inner mitochondrial membrane. The total number of mRNA species with annotated mitochondrial sublocalization is indicated above each column. See **Supplementary file 4A**. (C) Scheme for identifying putative RNAs associated with the nuclear lamina. Since subsets of laminar proteins should be biotinylated both by APEX2-NLS (*top right*) and by HRP-KDEL (*bottom right* and **Figure 3A**), we can intersect these two datasets to obtain a candidate list of nuclear lamina-localized RNAs. Notation as in **Figure 3A**. (D) Venn diagram identifying putative lamina-associated RNAs, defined as the overlap between HRP-KDEL- and APEX2-NLS-enriched RNAs. See also: **Supplementary file 4B**. Significance testing: hypergeometric test.

DOI: https://doi.org/10.7554/eLife.29224.013

mRNAs that encode secretory proteins (for which our quantification may convolve nuclear-retained pre-mRNAs and mature ER-localized transcripts), we observed 104 candidate laminar RNAs, including 48 mRNAs and 56 noncoding RNA species (*Figure 5D*; *Supplementary file 4B*). Although some portion of this highly speculative list may comprise experimental noise, the target RNAs identified here represent a compelling starting point for exploration of regulatory RNAs that have long remained elusive. Furthermore, given the flexibility with which APEX-RIP can be applied in different subcellular compartments, we anticipate that this form of analysis could be widely used to generate novel hypotheses regarding RNA subcellular localization in a diverse range of cellular contexts.

## Discussion

Methods for mapping RNA subcellular localization are constrained by the limits of their spatiotemporal precision, the diversity of RNA species that they can simultaneously analyze, their generality across cell types and compartments, and their ease of use. We believe that APEX-RIP holds substantial advantages to existing imaging- and sequencing-based techniques with regard to many of these factors.

Compared to imaging-based technologies, APEX-RIP offers superior target throughput, ease of use, and less cellular perturbation. For example, although modern variants of FISH can achieve extremely high spatial precision—even enabling the visualization of individual RNA molecules (*Batish et al., 2011*) this technique requires the synthesis and testing of customized fluorescent probes for each transcript of interest, a cumbersome process that limits throughput (*Cabili et al., 2015*). A highly multiplexed FISH variant, MERFISH, substantially boosts throughput—enabling thousands of transcripts to be simultaneously visualized—but requires complex protocols for probe set design and imaging (*Chen et al., 2015*). An alternate approach, FISSEQ, achieves similar target depth without the need for gene-specific probes, but instead relies on customized instrumentation and a rococo process of in situ sequencing and imaging (*Lee et al., 2014*). Notably, without incorporating additional stains or markers, these imaging-based approaches provide little information regarding the local environment (i.e., proximal cellular compartments or features) near each RNA target. Furthermore, these techniques are perturbative in that they require extensively fixing and permeabilizing cells prior to data collection (up to several hours in 1–4% formaldehyde) which can destroy membranes and alter endogenous RNA localization (*Fox et al., 1985*). This latter issue can be circumvented through a variety of live-cell imaging techniques, but these require the implementation of customized reagents that limit throughput, and may even distort the localization of the RNA targets under enquiry (*Paige et al., 2011*; *Hocine et al., 2013*; *Nelles et al., 2016*). By contrast, APEX-RIP is unencumbered by many of these constraints. It does not require the development of target-specific expression constructs or probes; nor does it rely on specialized instrumentation. The protocol captures RNA localization in living cells without detergent or methanol treatment so that membranes and spatial relationships are preserved. The ensemble of RNA targets analyzed (and, for that matter, the array of RNA classes analyzed) is theoretically limited only by the library synthesis and sequencing protocols employed. Moreover, since APEX-RIP captures only RNAs proximal to or within a specific subcellular compartment, it offers greater information content than do its imaging-based alternatives.

Compared to fractionation-based technologies, APEX-RIP offers superior accuracy, ease of use, and general versatility. As illustrated in the nucleus and ER, our technique rivals or outperforms

conventional Fractionation-Seq with regard to both target specificity and recall, apparently circumventing the dual issues of target loss and off-target contamination that can plague such approaches (*Figures 2E–F; 4C–E*). We ascribe this performance boost to two principal factors. First, the high spatiotemporal precision afforded by in situ biotinylation (*Rhee et al., 2013*) allows us to efficiently isolate target material from contaminants that might be difficult to remove by classical fractionation, thereby improving specificity. Second, covalently coupling target RNAs to affinity-tagged proteins allows us to recover low-abundance or weakly affiliated transcripts that might otherwise be lost during biochemical enrichment, thereby improving target recall (*Figure 3—figure supplement 2B*). Perhaps more importantly, however, we have achieved these results in a variety of subcellular compartments using a common protocol, thus obviating the need to develop customized purification schemes for each compartment. This generality should enable APEX-RIP to access 'unpurifiable' subcellular compartments for which such purification schemes would be impossible. While a related technology, proximity-dependent ribosome profiling, exhibits similar versatility within diverse subcellular milieus (*Jan et al., 2014*), this approach is limited to actively translated mRNAs. It also requires biotin starvation prior to tagging, which is toxic to mammalian cells, and as such, prevents widespread application. As we have demonstrated, APEX-RIP can map diverse classes of noncoding RNA and quiescent mRNA (*Figure 4B*), and eschews toxic protocols that starve cells of essential nutrients for prolonged periods of time.

The APEX-RIP methodology does have notable limitations. Cells to be analyzed must express a recombinant construct, in contrast to FISH and Fractionation-Seq, which can be performed on genetically unmodified cells, or on intact tissues. Application of APEX-RIP in developing animals, or in situ within animal nervous systems—cases where RNA localization is known to play a crucial regulatory role—would require the generation of a transgenic organism, and may be hindered by the need to deliver BP, $H_2O_2$, and formaldehyde into intact tissue. APEX-RIP also appears to exhibit poorer spatial specificity in membrane-free subcellular regions, since targeting APEX2 to cytoplasmic face of the endoplasmic reticulum failed to enrich secretome mRNAs from cytosolically-localized transcripts (*Figure 3E*). However, since the ER membrane forms a convoluted network that occupies a substantial volume of the cytosol, it is unclear the degree to which this apparent lack of specificity might apply to other, more discrete subcellular milieus.

The APEX peroxidase used here has also previously been used to generate contrast for electron microscopy in fixed cells (*Martell et al., 2012*; *Lam et al., 2015*), and for spatially-resolved proteomic mapping in living cells (*Rhee et al., 2013*; *Hung et al., 2014*; *Loh et al., 2016*; *Han et al., 2017*; *Hung et al., 2017*; *Mick et al., 2015*). This study extends APEX to a new class of applications and to a new biopolymer. In principle, it should be possible to use a single APEX-expressing cell line to characterize a target subcellular compartment by electron microscopic, proteomic, and transcriptomic means. Related methods for proteomic mapping, such as BioID (*Roux et al., 2012*), lack this versatility, because the underlying chemistry is not as flexible as the one-electron oxidation reaction catalyzed by APEX.

We anticipate that the initial subcellular transcriptomic map presented in this work—probing the mitochondrial matrix, cytosol, nucleus, and ER membrane of HEK 293T cells—will serve as a valuable resource for cell biologists. Analysis of these data has already yielded potential insight into nuclear-retained mRNAs, cytosolic lncRNAs, putative lamina-localized RNAs, and genes that may be translated locally at mitochondria-endoplasmic reticulum junctions. Applying APEX-RIP to other subcellular compartments will further expand the depth and breadth of this map. Furthermore, given the high temporal resolution of APEX-RIP, we imagine that our technology might enable profiling of subcellular RNA pools in response to acute stimuli or drugs, or throughout stages of the cell cycle and development.

## Materials and methods

**Key resources table**

| Reagent type (species) or resource | Designation | Source or reference | Identifiers | Additional information |
|---|---|---|---|---|

*Continued on next page*

Continued

| Reagent type (species) or resource | Designation | Source or reference | Identifiers | Additional information |
|---|---|---|---|---|
| cell line (human) | HEK293T | ATCC | CRL3216; RRID: CVCL_0063 | |
| cell line (human) | mito-APEX2 (HEK293T) | this paper | | mito-BamHI-V5-APEX2 CMV promoter Mito is a 24-amino acid mitochondrial targeting sequence (MTS) derived from COX4. V5: GKPIPNPLLGLDST |
| cell line (human) | APEX2-NLS (HEK293T) | this paper | | NotI-V5-APEX2-EcoRI-3xNLS-NheI CMV promoter NLS: DPKKKRKV |
| cell line (human) | APEX2-NES (HEK293T) | PMID: 28441135 | | BstBI-FLAG-APEX2-NES-NheI CMV promoter NES: LQLPPLERLTLD |
| cell line (human) | ERM-APEX2 (HEK293T) | PMID: 28441135 | | BstBI-ERM-APEX2-V5-NheI CMV promoter ERM is ER membrane targeting sequence derived from N-terminal 27 amino acids of rabbit P450 C1 (MDPVVVLGLCLSCLLLL SLWKQSYGGG) |
| cell line (human) | HRP-KDEL (HEK293T) | this paper | | NotI-IgK-HRP-V5-KDEL-IRES -puromycin-XbaI CMV promoter IgK is N-terminal signaling sequence that brings protein to ER (METDTLLLWVLLLWVPGSTGD). KDEL is ER-retaining sequence |
| antibody | Anti V5 | Life Technologies | R960-25; RRID: AB_2556564 | Dilution 1:1000 |
| antibody | Anti FLAG | Agilent | 200472 | Dilution 1:500 |
| antibody | Anti TOM20 | Santa Cruz Biotechnology | sc-11415; RRID: AB_2207533 | Dilution 1:400 |
| antibody | Anti RCN2 | Proteintech | 10193–2-AP; RRID: AB_2180018 | Dilution 1:200 |
| antibody | Anti Mouse-AlexaFlour488 | Life Technologies | A-11029; RRID: AB_2534088 | Dilution 1:1000 |
| antibody | Anti Mouse-AlexaFlour568 | Life Technologies | A-11031; RRID: AB_144696 | Dilution 1:1000 |
| antibody | Streptavidin-HRP | ThermoFisher | S-911 | Dilution 1:1000 |
| recombinant DNA reagent | Mito-APEX (plasmid) | PMID: 23371551 | pCDNA vector | |
| recombinant DNA reagent | mito-APEX2 (plasmid) | this paper | pLX304 vector | mito-BamHI-V5-APEX2 CMV promoter Mito is a 24-amino acid mitochondrial targeting sequence (MTS) derived from COX4. V5: GKPIPNPLLGLDST |
| recombinant DNA reagent | APEX2-NLS (plasmid) | this paper | | NotI-V5-APEX2-EcoRI-3xNLS-NheI CMV promoter NLS: DPKKKRKV |
| recombinant DNA reagent | HRP-KDEL (plasmid) | this paper | | NotI-IgK-HRP-V5-KDEL-IRES -puromycin-XbaI CMV promoter IgK is N-terminal signaling sequence that brings protein to ER (METDTLLLWVLLLWVPGSTGD). KDEL is ER-retaining sequence |

*Continued*

| Reagent type (species) or resource | Designation | Source or reference | Identifiers | Additional information |
|---|---|---|---|---|
| sequence-based reagent | Ribo-Zero Gold rRNA removal kit (Illumina) | Illiumina | MRZG12324 | |
| sequence-based reagent | Truseq RNA sample preparation kit V2 | Illiumina | RS-122–2001 | |
| sequence-based reagent | MT-ND1 forward | this paper | | CACCTCTAGCCTAGCCGTTT |
| sequence-based reagent | MT-ND1 reverse | this paper | | CCGATCAGGGCGTAGTTTGA |
| sequence-based reagent | MT-ND2 forward | this paper | | CTTAAACTCCAGCACCACGAC |
| sequence-based reagent | MT-ND2 reverse | this paper | | AGCTTGTTTCAGGTGCGAGA |
| sequence-based reagent | MT-ND3 forward | this paper | | CCGCGTCCCTTTCTCCATAA |
| sequence-based reagent | MT-ND3 reverse | this paper | | AGGGCTCATGGTAGGGGTAA |
| sequence-based reagent | MT-ND4 forward | this paper | | ACAACACAATGGGGCTCACT |
| sequence-based reagent | MT-ND4 reverse | this paper | | CCGGTAATGATGTCGGGGTT |
| sequence-based reagent | MT-ND4L forward | this paper | | TCGCTCACACCTCATATCCTC |
| sequence-based reagent | MT-ND4L reverse | this paper | | AGGCGGCAAAGACTAGTATGG |
| sequence-based reagent | MT-ND5 forward | this paper | | TCCATTGTCGCATCCACCTT |
| sequence-based reagent | MT-ND5 reverse | this paper | | GGTTGTTTGGGTTGTGGCTC |
| sequence-based reagent | MT-ND6 forward | this paper | | GGGTTGAGGTCTTGGTGAGT |
| sequence-based reagent | MT-ND6 reverse | this paper | | ACCAATCCTACCTCCATCGC |
| sequence-based reagent | MT-CYTB forward | this paper | | TCTTGCACGAAACGGGATCA |
| sequence-based reagent | MT-CYTB reverse | this paper | | CGAGGGCGTCTTTGATTGTG |
| sequence-based reagent | MT-COX1 forward | this paper | | TCCTTATTCGAGCCGAGCTG |
| sequence-based reagent | MT-COX1 reverse | this paper | | ACAAATGCATGGGCTGTGAC |
| sequence-based reagent | MT-COX2 forward | this paper | | AACCAAACCACTTTCACCGC |
| sequence-based reagent | MT-COX2 reverse | this paper | | CGATGGGCATGAAACTGTGG |
| sequence-based reagent | MT-COX3 forward | this paper | | CTAATGACCTCCGGCCTAGC |
| sequence-based reagent | MT-COX3 reverse | this paper | | AGGCCTAGTATGAGGAGCGT |
| sequence-based reagent | MT-ATP6 forward | this paper | | TTCGCTTCATTCATTGCCCC |
| sequence-based reagent | MT-ATP6 reverse | this paper | | GGGTGGTGATTAGTCGGTTGT |
| sequence-based reagent | MT-ATP8 forward | this paper | | ACTACCACCTACCTCCCTCAC |
| sequence-based reagent | MT-ATP8 reverse | this paper | | GGCAATGAATGAAGCGAACAGA |
| sequence-based reagent | MT-RNR1 forward | this paper | | CATCCCCGTTCCAGTGAGTT |
| sequence-based reagent | MT-RNR1 reverse | this paper | | TGGCTAGGCTAAGCGTTTTGA |
| sequence-based reagent | MT-RNR2 forward | this paper | | CAGCCGCTATTAAAGGTTCGT |
| sequence-based reagent | MT-RNR2 reverse | this paper | | AAGGCGCTTTGTGAAGTAGG |
| sequence-based reagent | GAPDH forward | this paper | | TTCGACAGTCAGCCGCATCTTCTT |
| sequence-based reagent | GAPDH reverse | this paper | | GCCCAATACGACCAAATCCGTTGA |
| sequence-based reagent | XIST forward | this paper | | CCCTACTAGCTCCTCGGACA |
| sequence-based reagent | XIST reverse | this paper | | ACACATGCAGCGTGGTATCT |
| sequence-based reagent | EMC10 forward | this paper | | TTCATTGAGCGCCTGGAGAT |
| sequence-based reagent | EMC10 reverse | this paper | | TTCATTGAGCGCCTGGAGAT |
| sequence-based reagent | PCSK1N forward | this paper | | GAGACACCCGACGTGGAC |
| sequence-based reagent | PCSK1N reverse | this paper | | AATCCGTCCCAGCAAGTACC |
| sequence-based reagent | SSR2 forward | this paper | | GTTTGGGATGCCAACGATGAG |
| sequence-based reagent | SSR2 reverse | this paper | | CTCCACGGCGTATCTGTTCA |

*Continued on next page*

*Continued*

| Reagent type (species) or resource | Designation | Source or reference | Identifiers | Additional information |
| --- | --- | --- | --- | --- |
| sequence-based reagent | TMX1 forward | this paper | | ACGGACGAGAACTGGAGAGA |
| sequence-based reagent | TMX1 reverse | this paper | | ATTTTGACAAGCAGGGCACC |
| sequence-based reagent | SFT2D2 forward | this paper | | CCATCTTCCTCATGGGACCAG |
| sequence-based reagent | SFT2D2 reverse | this paper | | GCAGAACACAGGGTAAGTGC |
| sequence-based reagent | EPT1 forward | this paper | | TGGCTTTCTGCTGGTCGTAT |
| sequence-based reagent | EPT1 reverse | this paper | | AATCCAAACCCAGTCAGGCA |
| sequence-based reagent | DRAP1 forward | this paper | | ACATCCCACCTGAAGCAGTG |
| sequence-based reagent | DRAP1 reverse | this paper | | GATGCCACCAGGTCCTTCAA |
| sequence-based reagent | FAU forward | this paper | | TCCTAAGGTGGCCAAACAGG |
| sequence-based reagent | FAU reverse | this paper | | GTGGGCACAACGTTGACAAA |
| sequence-based reagent | SUB1 forward | this paper | | CGTCACTTCCGGTTCTCTGT |
| sequence-based reagent | SUB1 reverse | this paper | | TGATTTAGGCATCGCTTCGC |
| sequence-based reagent | LSM6 forward | this paper | | CGGACGACCAGTTGTGGTAA |
| sequence-based reagent | LSM6 reverse | this paper | | CCAGGACCCCTCGATAATCC |
| sequence-based reagent | COPS2 forward | this paper | | AGGAGGACTACGACCTGGAAT |
| sequence-based reagent | COPS2 reverse | this paper | | GCCGCTTTTGGGTCATCTTC |
| sequence-based reagent | CGGBP1 forward | this paper | | GCCTCGTCCACTTTCCCTAA |
| sequence-based reagent | CGGBP1 reverse | this paper | | TCATGCCTTTACGTAGGATCGAG |
| sequence-based reagent | BCA53 forward | this paper | | TCTTGCCTGCTCCACAGTTT |
| sequence-based reagent | BCA53 reverse | this paper | | CAAACACCAAGGAGGGGTCT |
| sequence-based reagent | CEP128 forward | this paper | | TACAGTAATGGACAGGCGGG |
| sequence-based reagent | CEP128 reverse | this paper | | TCCGGAGTTGGTCGATTGAT |
| sequence-based reagent | MAD1L1 forward | this paper | | CGAGTCTGCCATCGTCCAA |
| sequence-based reagent | MAD1L1 reverse | this paper | | GCACTCTCCACCTGCTTCTT |
| sequence-based reagent | RAD51B forward | this paper | | TTTGGACGAAGCCCTGCAT |
| sequence-based reagent | RAD51B reverse | this paper | | CACAACCTGGTGGACCTGTA |
| sequence-based reagent | RBPMS forward | this paper | | ACAGTCGCTCAGAAGCAGAG |
| sequence-based reagent | RBPMS reverse | this paper | | CGAAGCGGATGCCATTCAAA |
| sequence-based reagent | TCF7 forward | this paper | | TCAACAGCCCACATCCCAC |
| sequence-based reagent | TCF7 reverse | this paper | | AGAGGCCTGTGAACTTGCTT |
| sequence-based reagent | HOOK2 forward | this paper | | TTTGCTGAAAAGGAAGCTGGA |
| sequence-based reagent | HOOK2 reverse | this paper | | GCAACTCCAGATCTGCCTCA |
| sequence-based reagent | MAN2C1 forward | this paper | | ATGAGGCCCACAAGTTCCTG |
| sequence-based reagent | MAN2C1 reverse | this paper | | TCTCATAGGTGGCCTGGGAA |
| peptide, recombinant protein | | | | |
| commercial assay or kit | | | | |
| chemical compound, drug | Biotin-phenol (BP) | PMID: 23371551 | | |
| software, algorithm | Tophat v2.1.1 | DOI: 10.1186/gb-2013-14-4-r36 | RRID:SCR_013035 | |
| software, algorithm | CuffDiff2 | | RRID:SCR_001647 | |
| software, algorithm | Slidebook 6.0 | | RRID:SCR_014300 | |
| software, algorithm | DAVID bioinformatics analysis | | RRID:SCR_003033 | |
| other | | | | |

## Plasmids and cloning

The pCDNA3 mito-APEX plasmid was published previously (*Rhee et al., 2013*). The Mito-APEX2 construct was cloned from this plasmid using a two-step protocol. First, the A134P mutation (*Lam et al., 2015*) was introduced into the APEX gene itself, using QuikChange mutagenesis (Agilent Technologies, Santa Clara, CA), and thereafter the APEX2 gene was moved to the lentiviral vector pLX304, via Gateway cloning (ThermoFisher Scientific, Waltham, MA), to generate the plasmid pLX304 mito-APEX2. Other APEX-fusion constructs (pLX304 APEX2-NLS, pLX304 APEX2-NES, and pLX304 ERM-APEX2) were cloned by Gibson assembly (New England Biolabs, Ipswich, MA), using PCR to add targeting sequences and Gibson Assembly homology arms to the APEX2 gene, and joining the resulting insert into the pLX304 vector digested by *BstBI* and *NheI*. To clone HRP-KDEL, the HRP-KDEL-IRES-Puromycin cassette from HRP C (*Martell et al., 2016*), was PCR-amplified and cloned into pCDNA3 using *NotI* and *XbaI* sites. Targeting sequences and restriction sites for all constructs are listed in (*Supplementary file 5A*).

## Mammalian cell culture

Human embryonic kidney (HEK) 293 T cells (RRID: CVCL_0063) were obtained, authenticated by STR profiling from ATCC, and cultured in growth media consisting of 1:1 DMEM:MEM (Cellgro, Thermo-Fisher Scientific, Manassas, VA), supplemented with 10% Fetal Bovine Serum (FBS), 50 units/mL penicillin, and 50 µg/mL streptomycin, at 37°C and under 5% $CO_2$. Cells were discarded at 25 passages. Cell lines were not tested for Mycoplasma contamination. For fluorescence microscopy imaging experiments (*Figures 1B*, *2A* and *3C*, and *Figure 2—figure supplement 3C*), cells were grown on 7 × 7 mm glass coverslips in 48-well plates. To improve cell adherence, coverslips were pretreated with 50 µg/mL fibronectin (Millipore, Burlington, MA) for 20 min at 37°C and washed once with Dulbecco's phosphate-buffered saline (DPBS), pH 7.4. Cells used for generating lentivirus were grown on T25 plates, in MEM supplemented as above, at 37 °C under 5% $CO_2$.

## Preparation of cell lines stably expressing APEX-fusion constructs

To prepare lentivirus, one ~ 70% confluent T25 plate of HEK 293T cells, grown as above, was co-transfected with 2.5 µg of APEX2 fusion plasmid, along with 0.25 µg and 2.25 µg, respectively, of the lentivirus packaging plasmids VSV-G, and dR8.91 (*Pagliarini et al., 2008*). Transfection mixes used 10 µL Lipofectamine 2000 (ThermoFisher Scientific) and were brought to a final volume of 2 mL with unsupplemented MEM. The cells were transfected for 3 hr, after which media was replaced with 2 ml of fresh growth media with FBS. After 48 hr, the lentiviral supernatant was collected by aspiration and filtered through a 0.45 µm syringe-mounted filter. This filtered supernatant was immediately used to infect cells. HEK293T cells, grown in 6-well plates as described above, were infected at ~50% confluency, grown for 2 days, followed by selection in growth medium supplemented with 8 µg/mL blasticidin for 7 days, before further analysis.

For the cells stably expressing HRP-KDEL, HEK293T cells at ~60% confluency, grown in 6-well plates as described above, were transfected with the mixture of 150 µg of plasmid and 10 µL Lipofectamine 2000 in unsupplemented MEM for 3 hr, after which media was replaced with 2 ml of fresh growth media with FBS. After 48 hr, the cells were trypsinized and replated in T25 flask in growth medium supplemented with 1 µg/mL puromycin for 7 days, before further analysis.

## Immunofluorescence staining and microscopy

For immunofluorescence experiments (*Figures 1B*, *2A* and *3C*, and *Figure 2—figure supplement 3C*), stable APEX- or HRP-expressing cells were BP-labeled and crosslinked, as described below, and subsequently fixed with 4% (v/v) paraformaldehyde in PBS at room temperature for 10 min. Cells were then washed with PBS three times and permeabilized with cold methanol at –20°C for 5 min. Cells were washed again three times with room-temperature PBS and then incubated with primary antibodies in PBS–supplemented with 1% (w/v) Bovine Serum Albumin (BSA)–for 1 hr at room temperature. After washing three times with PBS, cells were incubated with secondary antibodies and neutravidin-AlexaFluor647 (1:1000 dilution) in BSA-supplemented PBS for 30 min. Cells were then washed three times with PBS and imaged by confocal fluorescence microscopy, or in PBS at 4°C in light-tight containers prior to imaging. Primary and secondary antibodies used were listed in *Supplementary file 5B*.

Fluorescence confocal microscopy was performed with a Zeiss AxioObserver microscope with 63 × oil immersion objectives, outfitted with a Yokogawa spinning disk confocal head, a Cascade II:512 camera, a Quad-band notch dichroic mirror (405/488/568/647), 405 (diode), 491 (DPSS), 561 (DPSS) and 640 nm (diode) lasers (all 50 mW). Alexa Fluor488 (491 laser excitation, 528/38 emission), Alexa Fluor 568 (561 laser excitation, 617/73 emission), and AlexaFluor647 (640 laser excitation, 700/75 emission) and differential interference contrast (DIC) images were acquired through a 63x oil-immersion lens. Acquisition times ranged from 100 to 1,000 ms. For image acquisition and analysis, we used the SlideBook 6.0 software (Intelligent Imaging Innovations, Denver, CO, RRID:SCR_014300).

Unless otherwise noted, imaging data are representative of three independent experiments with ≥5 fields of view each.

## Immunofluorescence measuring biotinylated protein diffusion

HEK 293 T cells stably expressing APEX2-NLS were seeded onto fibronectin-coated coverslips and grown in 48-well plates, in 200 µL of 1:1 MEM:DMEM, supplemented with 15% (v/v) FBS, per well. At ~60% confluency, cells were transfected with a GFP expression plasmid (pCMV-EGFP, addgene plasmid 3525) using polyethyleneimine (PEI). Briefly, 150 ng plasmid was diluted into a 1:1 MEM: DMEM solution and incubated with 1 uL of PEI in a final reaction volume of 20 µL, for 15 min at room temperature, and added dropwise to cells. After 16 hr, cells were labeled and crosslinked according to BP–quench–then–crosslinking protocol (*see below*). At the indicated time points (*Figure 2—figure supplement 3C*), cell growth media was aspirated, and cells were fixed with 4% (v/v) formaldehyde in PBS supplemented, with 5 mM Trolox, 10 mM Ascorbate, 10 mM sodium azide, for 10 min at room temperature. Cells were washed twice with PBS, permeablized with methanol at −20°C for 5 min, and immunostained as described above. To stain the nucleus and biotinylated species, 0.1 ug/mL DAPI (4', 6-Diamidino-2-Phenylindole) and neutravidin-AlexaFluor647 (1:1000 dilution) were supplemented during the secondary antibody incubation. All primary and secondary antibodies used are listed in *Supplementary file 5B*. The data in *Figure 2—figure supplement 3C* are representative of the experiments with ≥15 fields of view each.

The nuclear and cytosolic biotinylation ratio (*Figure 2—figure supplement 3C*) was quantified using Slidebook 6.0. Nuclear biotinylation was quantified as the signal within the DAPI-stained area; cytosolic biotinylation was quantified as the signal within the GFP-labeled area, excluding that within DAPI-stained area.

## Western and streptavidin blotting

For blotting experiments (*Figures 1C* and *3D* and *Figure 2—figure supplement 1*), stable APEX- or HRP-expressing cells were grown in 6-well plates, as described above. After APEX labeling (*see below*), the cells were harvested by scraping, pelleted by centrifugation at 3,000 × g for 10 min, and stored at –80°C prior to use. Thawed pellets were lysed by gentle pipetting in RIPA lysis buffer (50 mM Tris, 150 mM NaCl, 0.1% SDS, 0.5% sodium deoxycholate, 1% Triton X-100, 5 mM EDTA), supplemented with 1 × protease cocktail (Sigma Aldrich, St Louis, MO), 1 mM PMSF (phenylmethylsulfonyl fluoride), for 5 min at 4°C. Lysates were then clarified by centrifugation at 15,000 × g for 10 min at 4°C before separation on homemade 8% SDS-PAGE gels. Gels were transferred to nitrocellulose membranes, stained by Ponceau S (0.1% (w/v) Ponceau S, 5% (v/v) acetic acid, in water) for 10 min at room temperature, and imaged. The blots were then blocked with blocking buffer (3% (w/v) BSA, 0.1% (v/v) Tween-20 in Tris-buffered saline) for 1 hr at room temperature, and incubated with primary antibodies in blocking buffer for 1 hr more. The dilutions of the antibodies are as followed: Mouse anti-V5 antibody (ThermoFisher Scientific RRID: AB_2556564) 1:1000 dilution and Mouse anti-FLAG antibody (ThermoFisher Scientific) 1:800 dilution. Blots were rinsed four times for 5 min with wash buffer (0.1% Tween-20 in Tris-buffered saline), and then immersed in blocking buffer supplemented with Goat anti-Mouse IgG H + L HRP Conjugate (1:3000 dilution, Bio-Rad Laboratories, Hercules, CA), for 1 hr at room temperature. Blots were rinsed four times for 5 min with wash buffer, and developed with the Clarity reagent (Bio-Rad Laboratories) and imaged on an Alpha Innotech gel imaging system. Processing of streptavidin blots was similar. Following Ponceau imaging, blots were blocked in blocking buffer for 30 min at room temperature, immersed in blocking buffer supplemented with streptavidin-HRP (1:3000 dilution, ThermoFisher Scientific, RRID:AB_2619743) at room

temperature for 15 min, rinsed with blocking buffer five times for 5 min each, developed and imaged using the Clarity reagent and an Alpha Innotech gel imaging system.

The data in these experiments (*Figures 1C* and *3D* and *Figure 2—figure supplement 1*) were also reproduced for quality control prior to quantitative PCR and sequencing.

## Quantitative RT–PCR

For quantitative RT–PCR (qRT–PCR, *Figure 1—figure supplement 1A,C*, *Figure 3E*, and *Figure 3— figure supplement 1A*) RNA samples (isolated as described below) were reverse transcribed using the SuperScript III Reverse Transcriptase kit (ThermoFisher Scientific), priming with random hexamers (ThermoFisher Scientific) according to the manufacturer's protocol. Samples were diluted with water, mixed with gene specific primers (*Supplementary file 5C*), and Rox-normalized FastStart Universal SYBR Green Master Mix (Roche Applied Sciences, Penzberg, Germany), and aliquotted into 384-well plates. qRT–PCR was performed on an Applied Biosystems 7900HT Fast real time PCR instrument, in quadruplicate. All threshold cycles ($C_t$, calculated per well) and efficiencies ($\varepsilon$, calculated per primer pair), were calculated from 'clipped' data, using Real time qPCR Miner (*Zhao and Fernald, 2005*). Primer pairs with average efficiencies below 90%—measured by qPCR Miner in at least three biological replicates, four technical replicates each—were omitted from further use. Raw $C_t$ values were corrected to account for the differences in sample volume, and percent yields were calculated via the ΔCt method:

$$yield = 100 \times (1 + \varepsilon)^{C_t}$$

…where in, $\Delta C_t = C_{t_{input}}\ corr - C_{t_{RIP}}\ corr$

Experimental uncertainties were calculated as described previously (*Shechner et al., 2015*). Given D = A–B, uncertainly was calculated using the formula:

$$\sigma_D = \sqrt{(\sigma_A)^2 + (\sigma_B)^2}$$

…wherein $\sigma_A$ and $\sigma_B$ are the measurement errors of A and B, respectively. For P, the product or quotient of values A and B, uncertainty was calculated using the formula:

$$\sigma_P = P \times \sqrt{\left(\frac{\sigma_A}{A}\right)^2 + \left(\frac{\sigma_B}{B}\right)^2}$$

The uncertainties of other functions, *f(x)*, were calculated using the first derivative approximation:

$$\sigma_{f(x)} = \sigma_x \times f'(x)$$

Sample sizes were determined in accordance with standard practices used in similar experiments in the literature; no sample-size estimates were performed to ensure adequate power to detect a prespecified effect size. Experiments were neither randomized nor blinded to experimental conditions. Each samples contained four technical replicates and no samples were excluded from analysis. Significance testing: Student's two-tailed t-test.

## APEX-RIP, Part I: optimized in situ biotinylation and crosslinking

Stable-expression HEK 293T cells were grown to 90% confluency in 6-well plates, as described above. Cells were incubated in fresh growth media supplemented with 500 µM Biotin Phenol (BP) (*Rhee et al., 2013*); also available from Iris Biotech GmbH, Marktredwitz, Germany) for 30 min at 37°C, after which cells were moved to room temperature and $H_2O_2$ was added to a final concentration of 1 mM. After 1 min, media was aspirated, and the APEX labeling reaction was quenched by addition of 2 mL azide-free quenching solution (10 mM ascorbate and 5 mM Trolox, in PBS), and further incubation at room temperature for 1 min. Thereafter, the liquid phase was aspirated, and cells were crosslinked by addition of 5 mL crosslink-quench solution (0.1% (v/v) formaldehyde, 10 mM sodium ascorbate, and 5 mM Trolox, in PBS). After 1 min, media were aspirated, and cells were again incubated in 5 mL fresh crosslink-quench solution, for 9 min at room temperature, with gentle agitation. The crosslinking reaction was terminated by addition of glycine (1.2 M stock, in PBS) to a final concentration of 125 mM, and gentle agitation for 5 min at room temperature. Cells were

washed twice with 2 mL room-temperature PBS, harvested by scraping, pelleted by centrifugation, and either processed immediately or flash frozen in liquid nitrogen and stored at –80°C before further analysis.

## APEX-RIP, Part II: Cell lysis, streptavidin bead enrichment of biotinylated material and RNA isolation

Unless otherwise noted, all buffers used during RNA isolation were supplemented to 0.1 U/ µL RNaseOUT (ThermoFisher Scientific), 1 × EDTA free proteinase inhibitor cocktail (ThermoFisher Scientific) and 0.5 mM DTT, final. Labeled, crosslinked cell pellets were thawed on ice (when necessary), and lysed by incubation in 1 mL ice-cold RIPA buffer, supplemented with 10 mM ascorbate and 5 mM Trolox, for 5 min at 4°C with end-over-end agitation. Samples were then sheared as described previously (*Hendrickson et al., 2016*) using a Branson Digital Sonifier 250 (Emerson Industrial Automation, St. Louis, MO) at 10% amplitude for three 30 s intervals (0.7 s on +1.3 s off), with 30 s resting steps between intervals. Samples were held in ice-cold metal thermal blocks throughout sonication. Lysates were then clarified by centrifugation at 15,000 × *g* for 5 min at 4°C, moved to fresh tubes and each diluted with 1 mL Native lysis buffer (NLB: 25 mM Tris pH 7.4, 150 mM KCl, 0.5% NP-40, 5 mM EDTA), supplemented with ascorbate and trolox. For each sample, 20% was removed as 'input;' to the remainder was added 50 µL of streptavidin-coated magnetic bead slurry (ThermoFisher Scientific ) that had been equilibrated by two washes in 1:1 RIPA: NLB. Samples were incubated for 2 hr at 4°C with end-over-end agitation. Beads were subsequently washed with the following series of buffers (1 mL each, 5 min per wash, 4°C, with gentle end-over-end agitation): (1) RIPA buffer, supplemented with trolox and ascorbate, (2) RIPA buffer without radical quenchers, (3) high salt buffer (1 M KCl, 50 mM Tris, pH 8.0, 5 mM EDTA), (4) urea buffer (2 M Urea, 50 mM Tris, pH 8.0, 5 mM EDTA), (5) RIPA Buffer, (6) 1:1 RIPA: NLB, (7) NLB, and (8) TE (10 mM Tris, pH 7.4, 1 mM EDTA).

Enriched RNAs were released from the beads by proteolysis in 100 µL of Elution Buffer (2% N-lauryl sarcoside, 10 mM EDTA, 5 mM DTT, in 1X PBS, supplemented with 200 µg proteinase K (ThermoFisher Scientific) and 4 U RNaseOUT) at 42°C for 1 hr, followed by 55°C for 1 hr, as previously described (*Hendrickson et al., 2016*). Eluted samples were cleaned up using Agencourt RNA-Clean XP magnetic beads (Beckman Coulter, Pasadena, CA), following the manufacturer's 1.5 mL tube format protocol, and eluted into 85 µL $H_2O$. Thereafter, contaminating DNA was removed by digestion with 5 U RQ1 RNase-free DNase I (Promega, Fitchburg, WI) in 100 µL of the manufacturer's supplied buffer (1X final concentration) at 37°C for 30 min. Purified RNAs were again cleaned up using Agencourt RNAClean XP magnetic beads, as above, and eluted into 30 µL $H_2O$. The concentration and integrity of all samples was measured using an Agilent 2100 Bioanalyzer, following the 'RNA Nano' or 'RNA Pico' protocols, where appropriate. Samples were not heat-cooled prior to loading Bioanalyzer chips.

## Alternate APEX-RIP biotinylation and crosslinking protocols

For Mito-APEX2 experiments (*Figure 1*), we followed a BP–then–crosslinking protocol that omitted the discrete radical quenching step (*Figure 1—figure supplement 1A*, *bottom*). Briefly, cells were grown and APEX-labeled as described above. Following the 1 min incubation in $H_2O_2$, cells were immediately treated with 5 mL crosslink-quench solution for one minute at room temperature, to simultaneously quench the APEX2 BP labeling reaction and initiate formaldehyde crosslinking. The liquid phase was aspirated, and cells were incubated in 5 mL of fresh crosslink-quench for two additional 1 min incubation steps, followed by a third, 8 min incubation at room temperature with gentle agitation.

Thereafter, crosslinking was terminated by the addition of glycine, and cells were harvested as described above. All subsequent steps (streptavidin bead enrichment, library prep, etc) proceeded as described above.

For the crosslinking–then–BP biotinylation protocol (*Figure 1—figure supplement 1A*, *top*), cells were washed once with 5 mL PBS, and crosslinked in 5 mL 0.1% (v/v) formaldehyde in PBS for 10 min at room temperature, with gentle agitation. The crosslinking reaction was quenched by addition of glycine (1.2 M, in PBS) to final concentration 125 mM, and gentle agitation for 5 min at room temperature. Crosslinked cells were then washed three times with PBS and incubated with 500 µM

biotin-phenol (BP) in PBS at room temperature, for 30 min. Thereafter, $H_2O_2$ was added to a final concentration 1 mM, for 1 min. The liquid phase was then removed by aspiration, and cells were washed twice with 2 mL quenching solution (5 mM Trolox, 10 mM Sodium Ascorbate, 10 mM sodium azide, in PBS). Crosslinked, labeled cells were harvested by scraping, and processed as described above.

## APEX-RIP, Part III: Library preparation, sequencing, and quantification

Purified RNA samples were depleted of ribosomal RNA using the Ribo-Zero Gold rRNA removal kit (Illumina, San Diego, CA), generally in accordance with the manufacturer's protocol. Briefly, RNA concentration and integrity were quantified on an Agilent 2100 Bioanalyzer, using 'RNA Pico' and, where appropriate, 'RNA Nano' kits. Samples were not concentrated prior to rRNA depletion, which can accommodate a maximum input volume of 17 µL. Therefore, samples with total input masses of $\leq$ 20 ng or 20–100 ng were mixed with 1 µL or 2 µL of Ribo-Zero rRNA Removal Solution, respectively, in 1x RiboZero Reaction Buffer, at a final volume of 20 µL. Reaction mixes were incubated at 68°C for 10 min, followed by 25°C for 5 min more, and thereafter added to 32.5 µL magnetic beads (90 µL bead slurry; washed with water and equilibrated in Magnetic Bead Resuspension Buffer, supplemented with RiboGuard RNase Inhibitor) by extensive pipetting. Binding reactions were incubated at room temperature for 5 min, gently vortexed for 5 s, and incubated for 5 min at 50°C, in a thermocycler. The supernatant, containing rRNA-depleted RNA, was diluted in water to 50 µL final volume, cleaned up with 50 µL Agencourt RNAClean XP beads and eluted with 19.5 µL of Elute, Prime, Fragment mix from the TruSeq RNA sample preparation kit, v2 (Illumina). Thereafter, libraries were prepared using the TruSeq RNA sample preparation kit, according to the manufacturer's instructions, starting from 'Incubate RFP' step. Each library was given a unique index during synthesis. Library concentration was measured, and quality confirmed, on an Agilent 2100 Bioanalyzer, using 'DNA High Sensitivity' kits.

While we did not explicitly include an RNA size-selection step in our library syntheses, we anticipate that smaller RNAs (tRNAs, snoRNAs, etc) would be relatively undersampled during our workflow. The mixing ratios used at all Agencourt bead-based cleanup steps (i.e. after reverse-crosslinking, during rRNA depletion, and throughout the early steps of library synthesis) disfavor the binding of such smaller species. For tRNAs, compact structure and post-transcriptional modifications can hinder amplification, making absolute quantification difficult (*Zheng et al., 2015*). Finally, the RNA fragmentation and library amplification steps have been optimized to generate libraries an average length of ~270 bp, as verified by BioAnalyzer. We assume that such undersampling applies equally to our input and RIP libraries, allowing us to compute fold enrichments, if not absolute abundances, for smaller RNAs that have somehow escaped de-enrichment (e.g. *Figure 1D–E*).

Indexed libraries were pooled in equimolar concentrations, with no more than ten libraries per pool, and subjected to 50 cycles of paired end sequencing, followed indexing, on two lanes of Illumina HiSeq 2500 flow cells, run in rapid mode (Genomics Core, Broad Institute of Harvard and MIT).

In general, three biological replicates for each construct were performed. Two biological replicates were performed for the mito-APEX experiment in *Figure 1—figure supplement 1B*.

As a basis of comparison, we here summarize the salient differences between our library preparation method, and those used in the alternative subcellular transcriptomics papers cited.

For the HEK 293T nuclear-cytoplasmic transcriptome datasets (*Sultan et al., 2014*), RNA isolation, library preparation and sequencing methods for the nuclear-cytoplasmic HEK293T dataset were generally similar to our own. Key differences include: (1) the analogous 'pre-enrichment' samples were obtained by Qiagen RNA extraction of live cells, (2) samples were not subjected to reverse-crosslinking or proteinase K treatment, and (3) following DNAse treatment, and RiboZero rRNA removal, samples were purified by ethanol precipitation with a glycogen carrier. Raw data were re-mapped and quantified in-house, using the same pipeline as was used for our own (*see below*).

Datasets for both ER Fractionation-Sequencing (*Reid and Nicchitta, 2012*) and proximity-restricted ribosome profiling (*Jan et al., 2014*) experiments were acquired by isolating ribosome-protected small RNA fragments, using methods that markedly differed from our own. In each case, fractionated and/or biotinylated polysomes were isolated and treated with RNAse. Monosome-protected RNA fragments were purified by gel electrophoresis, ligated to sequencing adaptors and reverse transcribed. Thereafter, Frac-Seq libraries were PCR amplified and subjected to SOLiD

sequencing; Ribosome profiling libraries were circularized before library amplification and subjected to single-end Illumina sequencing. We did not re-analyze data from these experiments: transcript quantifications were used as reported.

## Quantification of RNA abundances and folds enrichment; Assembly of true positive and false positive lists

Deep sequencing reads were mapped to human genome assembly hg19 using TopHat v2.1.1 (*Kim et al., 2013*), RRID:SCR_013035), with the flags, '–no-coverage-search' and '–GTF gencode. v19.annotation.gtf'. Gene expression was quantified against the Gencode v19 reference transcriptome (gencode.v19.annotation.gtf, genecodegenes.org) with Cufflinks v2.2.1. (*Trapnell et al., 2013*), RRID:SCR_014597), assessing the statistical significance of differential expression via CuffDiff2 (RRID:SCR_001647), with the flags, '–dispersion-method per-condition' and '–seed 42'.

No explicit filtering was imposed to mask the quantification of any RNA species: although nuclear-encoded tRNA, 5.8S, 18S, and 28S rRNA genes are absent from the Gencode reference transcriptome, and are hence opaque to our analysis, all other transcripts were quantified in an unbiased manner. Each RIP experiment was quantified independently. All Seq data will be made available through GEO under accession GSE106493.

Fold enrichments were calculated as follows:

$$log_2 \text{ fold change} = log_2 \left[ \frac{Average FPKM_{Post\ streptavidin\ enrichment}}{Average\ FPKM_{Pre\ streptavidin\ enrichment}} \right]$$

Significantly enriched genes in APEX-RIP, nuclear–cytosolic fractionation (*Sultan et al., 2014*), and ER-fractionation (*Reid and Nicchitta, 2012*) datasets were called as follows. RNAs with *p*-values greater than 0.05 (measured in CuffDiff, as described above) were removed from analysis. For ER-fractionation dataset (*Reid and Nicchitta, 2012*), RNAs with RPKM lower than 10 were filtered out. The remaining RNAs were then used to determine the enrichment threshold cutoffs, using Receiver Operating Characteristic (ROC) analysis (*Fawcett, 2006*), employing sets of true-positive and false-positive genes identified as described below. At each fold enrichment value, the true positive rate (TPR—the fraction of true positive genes identified as being enriched) and the false positive rate (FPR—the fraction of false positive genes identified as being enriched) were calculated. The fold enrichment value that maximizes the difference of these values (TPR–FPR) was chosen as the fold enrichment cutoff. In mitochondrial and ER-associated APEX-RIP experiments, ROC analysis was based on log_2 fold enrichment values comparing pre- and post-enrichment RNA abundances; in the nuclear-cytoplasmic experiment, it was based on calculated log_2 fold enrichment values comparing post-enrichment APEX2-NLS and APEX2-NES abundances.

The true and false positive gene sets needed for ROC analysis were defined as follows:

1. For mitochondrial APEX-RIP, true positives corresponded to the thirteen mitochondrial-encoded mRNAs; false positive RNAs corresponded to nuclear-encoded long non-coding RNAs.
2. For the nuclear and cytosolic partitioning experiment, the true positive list was defined as HEK293T-expressed long non-coding RNAs; the false positive list was the list of ER proximal RNAs (*Supplementary file 3D*) (*Jan et al., 2014*).
3. For ER-APEX-RIP, true positive genes were defined using data from ER-localized proximity-dependent ribosome profiling (*Jan et al., 2014*), applying a 'low-stringency' selection approach (*Supplementary file 3D*, 'Low-stringency ER list'). Namely, true-positives corresponded to all RNAs with input RPKM $\geq$5.0, input count $\geq$12, and log_2(fold enrichment)$\geq$ 0.904 (determined by ROC analysis) combined with all other HEK293T-expressed genes that were predicted by Phobius as having secretory signals, but which were absent from MitoCarta (*Pagliarini et al., 2008*). False positive RNAs were defined as all HEK293T-expressed genes lacking secretory signals, as predicted by Phobius (*Käll et al., 2004*), SignalP (*Petersen et al., 2011*), and TMHMM (*Krogh et al., 2001*).

## Coverage and specificity analysis of nuclear, cytosolic, and ER-proximal RNAs

To estimate the coverage (recall) and specificity of APEX-RIP at each subcellular compartment, we assembled lists of established target and off-target genes tailored for that compartment.

For analysis of the nuclear–cytosolic datasets (*Figure 2F*), our reference nuclear gene list comprised 827 lncRNAs with average RNA pre-enrichment abundances of 1.0 or greater. Our reference off-target list comprised the set of 1260 'Low-stringency' ER-proximal RNAs defined using proximity-restricted ribosome profiling (*Jan et al., 2014*), as described above (*Supplementary file 3D*, 'Low-stringency ER list').

For coverage analysis of the ER-proximal datasets (*Figure 4D*), our reference gene list comprised 71 mRNAs encoding ER-resident proteins (*Supplementary file 3E*). For specificity analysis (*Figure 4C,E*) a list of 'high-stringency' true positive genes (*Supplementary file 3D*, 'High-confidence ER list') was assembled using the ER-localized proximity-dependent ribosome profiling data (*Jan et al., 2014*), applying an input count cutoff of $\geq 100$ and a $\log_2$(fold enrichment) cutoff of $\geq 0.904$ (determined by ROC analysis, as above). The reference off-target list used in this analysis comprised 8855 mRNAs lacking secretory annotation, as assessed using Phobius, TMHMM, and SignalP, and which lacked the GOCC terms 'Endoplasmic reticulum,' 'Golgi,' 'membrane,' and 'extracellular' (*Ashburner et al., 2000*).

For analysis of contaminants in ER datasets (*Figure 3—figure supplement 2A*), the mRNAs that lacked predicted secretory annotation (assessed by Phobius, TMHMM, and SignalP, and by an absence of the GOCC terms 'Endoplasmic reticulum,' 'Golgi,' 'membrane,' and 'extracellular') were submitted to DAVID Bioinformatics analysis (*Huang et al., 2009*), RRID:SCR_003033). Only Gene Ontology terms that were enriched with *p*-values less than 0.05 —relative to the human background—are shown.

## Identification of candidate lamina-localized RNAs

To obtain an initial list of potential laminar RNAs, we identified transcripts that were significantly enriched both within the nucleus and near the ER membrane (*Figure 5C*). We manually curated our lists of APEX-RIP nuclear-localized and ER-associated RNAs (derived from ROC- and *p*-value analysis—*see above*—without further modification; *Supplementary files 2A* and *3A*), to identify transcripts that were significantly enriched in both. This resulted in a set of 441 overlapping RNAs (*Supplementary file 4B*), which we classified into transcript types according to standard GENCODE nomenclature. Statistically significant enrichment of overlapping RNAs in each class was calculated by hypergeometric test.

Of the initial 441 candidate RNAs, 337 correspond to mRNAs encoding secretory proteins, annotated as described above. However, since expression was measured at the gene level, and did not quantify individual RNA isoforms (*see above*) the apparent abundance of each gene stems from its mature and all immature (e.g. partially spliced) transcripts. Hence, the 337 secretory mRNAs in our overlapping set might be regarded as potential false positives, corresponding to cases where we have measured mature mRNAs near the ER surface, and partially processed precursor species in the nucleus, and not discrete species that reside at the interface of the nucleus and ER (i.e., the lamina). For this reason, we encourage omitting these genes in subsequent analysis of potential laminar RNAs (*Figure 5D*).

## Significance

RNA subcellular localization is a critical factor that influences a wide array of biological processes, ranging from *Drosophila* embryogenesis to mammalian neuronal signaling. However, while this spatial layer of transcriptome regulation has been characterized in a handful of contexts, a broader understanding of its overall extent, the factors governing its establishment, and its impact on biological function, remain inchoate. The limitations hindering this understanding have been largely technical, since conventional methods—such as fluorescence in situ hybridization (FISH) and Fractionation-Sequencing ('Frac-Seq')—depend upon specialized reagents and protocols that can limit throughput and general applicability. To address this fundamental need, we have developed a new strategy—APEX-RIP—which uses a simple toolkit and workflow to map the transcriptomes of discrete subcellular compartments at high depth and spatiotemporal resolution. APEX-RIP uses the engineered

ascorbate peroxidase APEX to biotinylate proteins within a target subcellular compartment in live cells; these affinity-tagged proteins are then chemically crosslinked in situ to nearby RNAs. When applied to a variety of membrane-enclosed and membrane-adjacent compartments, the APEX-RIP strategy exhibited target specificity and coverage rivaling or exceeding those attained by conventional fractionation-sequencing-based approaches, at a depth far exceeding those attainable by imaging-based methods. Furthermore, APEX-RIP can be applied to compartments that are recalcitrant to conventional biochemical purification. Given the superior precision, flexibility, and ease of this approach, we anticipate that APEX-RIP will provide a powerful tool for dissecting RNA subcellular localization in a broad range of biological contexts.

## Acknowledgements

We thank Jeffrey Martell for valuable experimental advice, Ozan Aygun for the curated ER protein list and assistance generating APEX2-NLS and HRP-KDEL stable cell lines, Chinmay Shukla and Furqan Fazal for valuable computational advice, and members of the Ting and Rinn labs for their constructive insights and critiques. Funding was provided by the NIH (R01-CA186568 to AYT and U01 DA040612 to JLR) and Stanford (to AYT).

## Additional information

### Funding

| Funder | Grant reference number | Author |
|---|---|---|
| National Institutes of Health | R01-CA186568 | Alice Y Ting |
| Stanford University | | Alice Y Ting |
| National Institutes of Health | U01-DA040612 | John L Rinn |

The funders had no role in study design, data collection and interpretation, or the decision to submit the work for publication.

### Author contributions

Pornchai Kaewsapsak, Conceptualization, Data curation, Validation, Investigation, Visualization, Methodology, Writing—original draft, Writing—review and editing; David Michael Shechner, Conceptualization, Validation, Investigation, Visualization, Methodology, Writing—original draft, Writing—review and editing; William Mallard, Formal analysis, RNA-Seq analysis; John L Rinn, Supervision, Funding acquisition, Writing—review and editing; Alice Y Ting, Conceptualization, Supervision, Funding acquisition, Visualization, Methodology, Writing—original draft, Writing—review and editing

### Author ORCIDs

Pornchai Kaewsapsak http://orcid.org/0000-0001-7921-0868
David Michael Shechner https://orcid.org/0000-0002-0574-256X
William Mallard http://orcid.org/0000-0002-2271-945X
John L Rinn https://orcid.org/0000-0002-7231-7539
Alice Y Ting https://orcid.org/0000-0002-8277-5226

### Decision letter and Author response

Decision letter https://doi.org/10.7554/eLife.29224.030
Author response https://doi.org/10.7554/eLife.29224.031

## Additional files

### Supplementary files

• Supplementary file 1. Mitochondrial APEX-RIP Data. (A) RNAs enriched by mito-APEX2-RIP. (B) Unfiltered mito-APEX-RIP RNA-Seq data. (C) Column definitions.

DOI: https://doi.org/10.7554/eLife.29224.014

• Supplementary file 2. Nuclear and Cytosolic APEX-RIP Data. (A) APEX2-RIP-enriched nuclear RNAs. (B) APEX2-RIP-enriched cytosolic RNAs. (C) Unfiltered APEX-RIP RNA-Seq data. (D) Unfiltered nuclear and cytosolic fractionation-seq data (*Sultan et al., 2014*). (E) Fractionation-Seq-enriched Nuclear RNAs. (F) Fractionation-Seq-enriched cytosolic RNAs. (G) lncRNAs used to determine coverage analysis (*Figure 2F*). (H) Column definitions.

DOI: https://doi.org/10.7554/eLife.29224.015

• Supplementary file 3. KDEL-RIP Data (ER-proximal RNAs). (A) KDEL-RIP-enriched ER-proximal RNAs. (B) Unfiltered KDEL-RIP RNA-Seq data. (C) ER-associated RNAs enriched by Fractionation-Seq (*Reid and Nicchitta, 2012*). (D) ER-associated RNAs enriched by proximity-dependent ribosome profiling (*Jan et al., 2014*). (E) True positive list: RNAs encoding established ER-resident proteins. (F) Column definitions.

DOI: https://doi.org/10.7554/eLife.29224.016

• Supplementary file 4. Additional analysis of ER and nuclear datasets. (A) Mitochondrial mRNAs (nuclear-encoded) enriched at the ER membrane. (B) RNAs that may be enriched at the nuclear lamina. (C) Column definitions.

DOI: https://doi.org/10.7554/eLife.29224.017

• Supplementary file 5. Materials used in this study. (A) Genetic constructs used in this study. (B) Antibodies used for immunofluorescence. RRID: Research Resource Identifier (https://scicrunch.org/resources). (C) qRT-PCR primers used in this study. (D) Column definitions.

DOI: https://doi.org/10.7554/eLife.29224.018

• Transparent reporting form

DOI: https://doi.org/10.7554/eLife.29224.019

## Major datasets

The following dataset was generated:

| Author(s) | Year | Dataset title | Dataset URL | Database, license, and accessibility information |
| --- | --- | --- | --- | --- |
| Pornchai Kaewsapsak, David Michael Shechner, William Mallard, John L Rinn, Alice Y Ting | 2017 | Live-cell mapping of organelle-associated RNAs via proximity biotinylation combined with protein-RNA crosslinking | https://www.ncbi.nlm.nih.gov/geo/query/acc.cgi?acc=GSE106493 | Gene Expression Omnibus (accession no: GSE106493) |

The following previously published datasets were used:

| Author(s) | Year | Dataset title | Dataset URL | Database, license, and accessibility information |
| --- | --- | --- | --- | --- |
| Sultan M, Amstislavskiy V, Risch T, Schuette M, Dökel S, Ralser M, Balzereit D, Lehrach H, Yaspo ML | 2014 | Influence of RNA extraction methods and library selection schemes on RNA-seq data | https://www.ebi.ac.uk/ena/data/view/PRJEB4197 | Publicly available at the EBI European Nucleotide Archive (accession no: PRJEB4197) |
| Jan CH, Williams CC, Weissman JS | 2014 | Principles of ER Co-Translational Translocation Revealed by Proximity-Specific Ribosome Profiling | https://www.ncbi.nlm.nih.gov/geo/query/acc.cgi?acc=GSE61012 | Gene Expression Omnibus (accession no: GSE61012) |
| Reid DW, Nicchitta CV | 2012 | Ribosome footprinting in the cytosol and endoplasmic reticulum | https://www.ncbi.nlm.nih.gov/geo/query/acc.cgi?acc=GSE31539 | Gene Expression Omnibus (accession no: GSE31539) |

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
