## [Decision Letter]

Thank you for submitting your article "Live-cell mapping of organelle-associated RNAs via proximity biotinylation combined with protein-RNA crosslinking" for consideration by *eLife*. Your article has been favorably evaluated by Vivek Malhotra (Senior Editor) and three reviewers, one of whom served as Guest Reviewing Editor. The reviewers have opted to remain anonymous.

The reviewers have discussed the reviews with one another and the Reviewing Editor has drafted this decision to help you prepare a revised submission.

This manuscript presents a new method, APEX-RIP, for detecting RNAs enriched in subcellular compartments. The method is an extension of this lab's previous method for isolation of protein complexes following in vivo proximity biotinylation using the engineered APEX2 enzyme. Here, APEX2 or HRP biotin labeling of proteins is combined with cross-linking and immunoprecipitation of RNA to probe the transcriptomes of specific subcellular locales. The method is not restricted to mRNAs, but can identify non-coding RNAs as well. The authors apply the method to map the RNA neighborhood compositions for cytosolic, nuclear, mitochondrial and ER cellular compartments in human HEK293 cells. The authors argue that the method advances the field of RNA localization and provides better temporal resolution, recall and specificity than current techniques including proximity-dependent ribosome profiling and subcellular fractionation-seq. To demonstrate the utility of the data, the authors offer interesting hypotheses about mitochondrial-ER junction and nuclear lamina RNAs, but these are highly speculative and not adequately supported by the data currently provided. There are also concerns about the spatial specificity and resolution of the method and the ability to discriminate contaminants from bona fide localization as well as the applicability to non-membranous compartments. Overall, the paper is well written, and data are presented that generally support the potential of APEX-RIP as a useful method to characterize subcellular transcriptomes. We have the following suggestions to better validate the method and improve the manuscript.

Essential revisions:

1) A major concern relates to the spatial specificity in membrane-free subcellular regions. The APEX-RIP methodology appears to work well for tightly bounded compartments (e.g. the inner mitochondrial matrix), but its specificity declines rapidly as compartments become less well defined. While the authors do mention this limitation in the Discussion, the message of the paper should be rebalanced towards clarifying these drawbacks. In all approaches the authors detect significantly more RNAs than previous methods and explain this solely with higher sensitivity of the approach without adequately validating this claim. In particular, they fail to address the possibility that a certain proportion of the hits may be false positives resulting from diffusion of biotinylated proteins prior to crosslinking. Contrary to what the authors state that capture of RNA does not happen "during a one-minute reaction". Instead, those proteins that are biotinylated during a one-minute reaction are in principle free to diffuse and interact with additional RNA molecules for several minutes until crosslinking is complete. For example, many nuclear RNA-binding proteins that are presumably targets of APEX2-NLS and between the nucleus and cytoplasm. Given that nuclear shuttling occurs on the msec time scale but there is a minute between treatment with peroxide and crosslinking, and that these relatively abundant proteins directly contact RNA, it seems likely that a significant fraction of the processed mRNAs found in the APEX2-NLS experiment may not be nuclear RNAs at all, but cytoplasmic RNAs crosslinked to RNA-binding proteins that were biotinylated in the nucleus but transited to the cytoplasm. Similarly, the authors find that targeting APEX/HRP to the lumen of the ER results in much higher specificity of RNA recovery than when it is targeted to the ER surface. Presumably this is connected to the more limited diffusion of proteins that reside in the ER lumen vs. the surface in the period immediately following biotin labeling.

Additionally, given the relatively mild and slow crosslinking conditions used, it seems inconsistent to assert that other RNA localization methods "fundamentally lack temporal precision [because] each requires extensively fixing and permeabilizing cells prior to data collection, during which time diffusion or the loss of cellular integrity can perturb endogenous RNA localization." The authors then state that "APEX-RIP is not encumbered by any of these constraints", but one could argue that during a 10 minute formaldehyde crosslinking procedure at room temperature, many biotinylated proteins are likely to keep diffusing and interacting with additional RNA molecules for some time. Have the authors considered alternatives to formaldehyde crosslinking e.g. UV crosslinking, which could be performed on ice in a shorter time, thus potentially limiting diffusion of biotinylated proteins?

2) Another potential limitation of the method that should at least be discussed is applicability of the method to animal level analysis, for example eggs, embryos, or the nervous system of an intact animal, where mRNA localization is a crucial regulatory mechanism.

3) Another major concern is discriminating contaminants as opposed to identifying novel subcellular RNA localization. Regarding the result (subsection “Hypotheses from ER and nuclear APEX-RIP datasets”, first paragraph) that 141 of the 2635 mRNAs in the ERM dataset code for mitochondrial proteins, it isn't clear why these were not identified as conspicuous "mitochondrion" contaminants when ontology analysis was done in the fractionation comparison. This analysis identified only 13 mitochondrion by ER fractionation-seq. The same might be said for the overlap of ER enriched and APEX-NLS enriched RNAs to arrive at the nuclear lamina localized RNA candidates. Why are these candidates not detected as contaminants in either individual set?

4) The novel hypotheses generated from the APEX-RIP data (mito-ER junction and nuclear lamina RNAs) are potentially interesting but in both cases highly speculative. Both ideas would be greatly strengthened if the authors could validate that some APEX-RIP candidate RNAs indeed localize to the suggested sites (e.g. by FISH or live RNA tracking).

5) The analysis of RNAs potentially associated with the nuclear lamina seems problematic. Whereas 14% of the set of nuclear RNAs detected overlap with the ER-associated RNAs, only 6% of the lncRNAs overlap. Thus, the lncRNAs are under-represented in the population of nuclear RNAs that could be associated with the lamina. The caveat that RNAs identified as nuclear may not actually be nuclear further weakens this analysis.

6) In the third paragraph of the Discussion, and at several other places of the manuscript, the authors make claims such as: "Compared to fractionation-based technologies, APEX-RIP offers superior accuracy". This seems like an over-interpretation based on the evidence presented, especially for the analyses of nucleo-cytoplasmic RNA distribution profiles. Firstly, the authors appear to draw this conclusion by comparing APEX-RIP and Frac-seq data from different cell lines (e.g. Figure 2). While it is likely that the nucleo-cytoplasmic localization of RNAs may exhibit a high degree of similarity between cell lines, comparison of results generated from the same cell lines would seem important to draw such conclusions. Second, the detection of ER mRNAs within the nucleus in Frac-seq data, but not APEX2-NLS samples, could simply underline the ability of the Frac-seq approach to purify and detect pre-mRNAs. Do the authors detect such species within APEX2-NLS RNA-seq dataset? Is read coverage of intronic regions comparable to that observed with nuclear Frac-seq?

7) The authors provide evidence that APEX-RIP captures RNAs with lower abundances than alternative approaches do, but it would be nice to have some sort of estimate of the limit of detection. Furthermore, there is no discussion for any of the experiments about false negatives.

8) Regarding RNAseq library preparation and data analysis, it would be useful to include more detail in both the Materials and methods section and the main text to demonstrate that their sequencing datasets and the different reference datasets are indeed directly comparable. Specifically, in the Materials and methods section the authors should include the dataset IDs for the specific ENCODE datasets and explain whether they as well as the other mentioned datasets were generated using comparable library preparation and sequencing methods. While the authors specifically mention lncRNAs, it is unclear how other, often highly abundant, non-protein coding RNAs were treated in the analysis. While this should be explained in detail in the Materials and methods, it would be useful to also include a brief description in the main text.

9) For IF assays to confirm proper APEX/HRP fusion subcellular targeting, co-labeling with organelle specific markers should be shown.

---

## [Author Response]

Essential revisions:1) A major concern relates to the spatial specificity in membrane-free subcellular regions. The APEX-RIP methodology appears to work well for tightly bounded compartments (e.g. the inner mitochondrial matrix), but its specificity declines rapidly as compartments become less well defined. While the authors do mention this limitation in the Discussion, the message of the paper should be rebalanced towards clarifying these drawbacks.

Inspired by this critique, we have edited the manuscript to rebalance the message of the paper. We now make clear at multiple points within the revised text (Introduction, Results, and Discussion) that APEX-RIP has drawbacks, most notably lower spatial specificity in compartments that are less well defined. However, we would like to note that APEX-RIP *was* able to successfully map the transcriptome of an “open” compartment (the cytosolic face of the ER), though doing so entailed biotinylating that organelle’s lumen. Given that many biological processes occur in membrane-proximal regions, we feel that this approach should still have broad utility and be helpful to a wide range of biologists.

In all approaches the authors detect significantly more RNAs than previous methods and explain this solely with higher sensitivity of the approach without adequately validating this claim.

There are two possible reasons for why we detect more RNAs than previous approaches. One is that we have higher sensitivity. Another is that we recover more non-specific hits (i.e., have more false positives). We believe that the data presented in this study support the first explanation rather than the second. We note:

1) New Figure 3—figure supplement 2 shows that our method is indeed more sensitive: APEX-RIP was able to recover established, truepositive RNAs of lower abundances than could either fractionation-Seq or proximity-dependent ribosome profiling.

2) For each compartment, we analyze the specificity of the resulting dataset in the best way that we know how. Figure 2; Figure 4 show that our specificity is not worse than other methods; in some cases, it is noticeably better. For example, our ER dataset is 96.5% specific (96.5% of enriched mRNAs encode transmembrane or secreted proteins), while ER fractionation-seq analyzed in an identical manner is 91% specific.

3) We also directly calculate sensitivity for each dataset, which we term “coverage”, or recall of expected RNAs. In Figure 4 for example, we have better recall/coverage/sensitivity than do either fractionation-seq or proximity-dependent ribosome profiling, analyzed in an identical fashion.

4) When we take a closer look at the RNAs that we recover that other methods do not (i.e., the non-overlapping part of the Venn diagram in Figure 4), we do not observe a substantial reduction in specificity, as measured by prior annotation. For example, for the ER, we detect 1281 RNAs that were missed by both Fractionation-Seq and proximity-dependent ribosome profiling. The specificity of mRNAs in this group is 93.4%, close to the specificity of the entire dataset of 2672 RNAs (of which 2494 are mRNAs). This strongly suggests that the additional RNAs we capture are still specific hits, and that we are not collecting more hits at the expense of specificity.

5) Comparing to ribosome profiling in particular, it makes sense that we should capture more, since we can enrich non-coding or silent mRNAs whereas their method cannot.

In particular, they fail to address the possibility that a certain proportion of the hits may be false positives resulting from diffusion of biotinylated proteins prior to crosslinking. Contrary to what the authors state that capture of RNA does not happen "during a one-minute reaction". Instead, those proteins that are biotinylated during a one-minute reaction are in principle free to diffuse and interact with additional RNA molecules for several minutes until crosslinking is complete. For example, many nuclear RNA-binding proteins that are presumably targets of APEX2-NLS and between the nucleus and cytoplasm. Given that nuclear shuttling occurs on the msec time scale but there is a minute between treatment with peroxide and crosslinking, and that these relatively abundant proteins directly contact RNA, it seems likely that a significant fraction of the processed mRNAs found in the APEX2-NLS experiment may not be nuclear RNAs at all, but cytoplasmic RNAs crosslinked to RNA-binding proteins that were biotinylated in the nucleus but transited to the cytoplasm.

We thank the reviewer for pointing out this very legitimate concern, which has prompted us to perform a new set of experiments, shown in new Figure 2—figure supplement 3 of the revised manuscript. Briefly, if a sizeable portion of biotinylated proteins-sufficient to distort our sequencing data-shuttle between the nucleus and cytosol during the “dead time” of our experiment, it should be possible to detect this redistribution by imaging. However, when monitoring the localization of biotinylated species during the course of the 17-minute biotinylation/formaldehyde crosslinking/quenching steps, we observed no significant loss of signal from within the compartment of origin (here, the nucleus) or concomitant increase in signal in the adjoining compartment (here, the cytosol; *p-*value, comparing time points = 0.374). From these data, we believe that it is unlikely that proteins biotinylated in the nucleus shuttle into the cytosol and give non-specific RNA capture to a significant degree. This claim is also supported by the RNA-Seq data itself, which shows minimal enrichment of cytosolic contaminants (e.g., ER-associated mRNAs from Jan et al.) in our nuclear APEX-RIP dataset.

Similarly, the authors find that targeting APEX/HRP to the lumen of the ER results in much higher specificity of RNA recovery than when it is targeted to the ER surface. Presumably this is connected to the more limited diffusion of proteins that reside in the ER lumen vs. the surface in the period immediately following biotin labeling.

We have edited the text to add this point. We also believe that the increased specificity results from the difference in protein biotinylation patterns resulting from HRP-KDEL versus APEX2-ERM labeling. Because the biotin-phenoxyl radical does not cross cellular membranes, HRP-KDEL biotinylation stops abruptly at the ER membrane. In contrast, APEX-ERM biotinylation forms a “contour map” in which protein biotinylation is strongest at the ER membrane, and falls off nanometer by nanometer into the cytosol, but is still detectable even tens of nanometers away. When performing mass spec proteomics, we address this by ratioing biotinylation extent against a cytosolic APEX2-NES reference construct. Here, for APEX-RIP, we believe the formaldehyde crosslinking step captures distal RNAs in the case of APEX2-ERM biotinylation, but captures strictly ERM-proximal RNAs in the case of HRP-KDEL, whose biotinylation products are restricted to the ER lumen.

Additionally, given the relatively mild and slow crosslinking conditions used, it seems inconsistent to assert that other RNA localization methods "fundamentally lack temporal precision [because] each requires extensively fixing and permeabilizing cells prior to data collection, during which time diffusion or the loss of cellular integrity can perturb endogenous RNA localization." The authors then state that "APEX-RIP is not encumbered by any of these constraints", but one could argue that during a 10 minute formaldehyde crosslinking procedure at room temperature, many biotinylated proteins are likely to keep diffusing and interacting with additional RNA molecules for some time.

This is a valid point and we have edited the text to make the comparison to other methods more accurate. We now state that, compared to imaging-based approaches, the main advantages of APEX-RIP are (1) no use of detergent/methanol to permeabilize membranes, which can disrupt spatial relationships, and (2) full sequence data instead of just gene IDs. Hence more information can be gleaned about RNAs in specific locales.

Have the authors considered alternatives to formaldehyde crosslinking e.g. UV crosslinking, which could be performed on ice in a shorter time, thus potentially limiting diffusion of biotinylated proteins?

We agree that APEX-CLIP is a great idea, and are interested in pursuing it in future studies. However, we have disfavored its use here for several reasons. Since APEX-catalyzed biotinylation preferentially labels the outer surface of RNPs, and since UV-induced crosslinks only form at the very cores of these complexes, we feared (though have not tested ourselves) that this disconnection between the points of labeling and contact might result in a substantial loss of signal in multi-subunit complexes. We were also concerned that various RNA•protein interactions might exhibit a range of different crosslinking efficiencies, thereby introducing bias into our sampling of the organelle-wide transcriptome (i.e. over-sampling RNAs that easily crosslinked, while undersampling those that aren’t). Although the power of CLIP represents an attractive alternative, we believe that exploring this alternative falls beyond the scope of the current study.

2) Another potential limitation of the method that should at least be discussed is applicability of the method to animal level analysis, for example eggs, embryos, or the nervous system of an intact animal, where mRNA localization is a crucial regulatory mechanism.

This is an excellent point. We agree that a limitation of using APEX-RIP in animals or tissue is that one must contend with BP/H2O2/formaldehyde perfusion into tissue. FISH and fractionation-Seq are more tissue-ready in this respect. We have edited the Discussion to point this out.

3) Another major concern is discriminating contaminants as opposed to identifying novel subcellular RNA localization. Regarding the result (subsection “Hypotheses from ER and nuclear APEX-RIP datasets”, first paragraph) that 141 of the 2635 mRNAs in the ERM dataset code for mitochondrial proteins, it isn't clear why these were not identified as conspicuous "mitochondrion" contaminants when ontology analysis was done in the fractionation comparison. This analysis identified only 13 mitochondrion by ER fractionation-seq.

The jury is still out on where nuclear-encoded, mitochondrial-resident proteins are translated. They may be translated in the open cytosol, near the outer mitochondrial membrane, at the ER membrane, or even at mito-ER contacts. Thus, for evaluating the specificity of our ERM dataset, we did not think it fair to automatically classify anything with mitochondrial annotation as a “contaminant”. Instead, for the specificity analysis in Figure 4 , our approach was to classify any mRNA encoding a transmembrane protein or secreted protein (according to Phobius, TMHMM, SignalP, and GOCC annotation) as “specific” as these are reasonable candidates for translation at the ER membrane. mRNAs lacking such annotation are potentially “non-specific” as noted in the bar graph.

Specifically for the 135 mRNAs (new number, based on new analysis pipeline that takes p-values into account; see Methods) in our ER dataset that code for mitochondrial proteins, 132 of these have transmembrane annotation. So we do not count them as “non-specific”. The remaining three mitochondrial genes comprise 3.4% of the total set of non-secretory/non-TM mRNAs enriched by KDEL-RIP (87 mRNAs total), which is why they aren’t evident in Figure 3—figure supplement 2 right. For comparison, of the 114 non-secretory/non-TM mRNAs enriched by fractionation-sequencing, 13 (11.4%) exhibit mitochondrial annotation.

The same might be said for the overlap of ER enriched and APEX-NLS enriched RNAs to arrive at the nuclear lamina localized RNA candidates. Why are these candidates not detected as contaminants in either individual set?

To address the valid critique, we will divide it into two parts: identifying contaminants in the nuclear dataset, and identifying contaminants in the ER KDEL-RIP dataset.

Finding contaminants in the Nuclear RNA experiment is difficult, largely because it’s hard to know, a priori, what those contaminants should be. All RNAs start off life in the nucleus, and so all expressed genes should be represented there, to some extent. Knowing this, we have approached the issue of specificity in our nuclear APEX-RIP dataset by two means. First, we looked for enrichment of lncRNAs in the nuclear dataset, and depletion of these transcripts in the cytosolic APEX-RIP data (Figure 2). Second, we looked for the depletion of mRNAs in the nuclear data, using the secretome mRNAs–a class of transcripts that should be predominantly cytoplasmic–as a proxy for mature mRNAs as a whole. In theory, we could try to expand that analysis–using *all* mature mRNAs as potential contaminants–to systematically address this issue. However, this analysis would require discrete quantification of the pre-processed and mature transcripts, for which we lack adequate sequencing depth. Given these considerations, we believe our approach – while imperfect – is reasonable.

The second part of the question pertains to why nuclear-localized RNAs are not classified as contaminants in the ER KDEL-RIP dataset. We have taken care to approach this issue as conservatively as possible in our analysis. For example, of the 441 RNAs that are enriched both in our nuclear and KDEL-RIP datasets, 385 are mRNAs, 337 (87.5%) of which have secretory/TM annotation. The skeptical interpretation is that the KDEL-RIP experiment has enriched mature secretory/TM mRNAs localized at the ER surface, while the NLS-RIP experiment has enriched their corresponding immature species within the nucleus. Hence, we have masked them from being potential laminar candidates.

Looking more broadly at the KDEL dataset: of the 2494 KDEL-enriched mRNAs, only 87 (3.5%) lack secretory/TM annotation – one might initially consider these few RNAs as potential contaminants. However, 48 of these transcripts (55%) are also enriched in the NLS dataset. By comparison, only 14% of KDEL-enriched secretome mRNAs (337, out of 2407 total) are enriched in the NLS-APEX-RIP, indicating that the putative KDEL “contaminant” list is highly enriched in nuclearlocalized RNA. Given the generally high specificity of our ER-associated RNA list, and the fact that these 48 are also nuclear-enriched, we have flagged them as being candidate laminar RNAs.

In the revised manuscript, we have attempted to clarify our writing so as to emphasize these points. We furthermore explicitly discuss the possibility that some of our novel hits may be ascribed to experimental noise.

4) The novel hypotheses generated from the APEX-RIP data (mito-ER junction and nuclear lamina RNAs) are potentially interesting but in both cases highly speculative. Both ideas would be greatly strengthened if the authors could validate that some APEX-RIP candidate RNAs indeed localize to the suggested sites (e.g. by FISH or live RNA tracking).

We believe that this request is beyond the scope of this work, which was submitted as a Tools and Resources article, and accordingly presents new methodology for studying RNA localization in living cells, and multiple high quality datasets that contain thousands of novel RNAs not previously associated with specific organelles (i.e., Resources). We parse the data to identify mito-ER and lamina candidates only to illustrate how the new datasets that we contribute can be mined in straightforward ways for novel hypotheses–hypotheses that are excellent starting points for new bodies of work.

Furthermore, we note that follow-up validation is not a simple matter that can be completed in 1-2 months. There are major technical considerations that suggest such follow up would be more suitable for a new study rather than as an addition to this work. First, the live RNA tracking methods suggested by the reviewer (using, for example, fluorescent dye-binding aptamers, or cassettes of fluorescently-labeled RNA-binding proteins) require either expressing the target RNA as a chimeric transgene – which might not necessarily recapitulate the expression regime and subcellular localization of the endogenous gene – or editing the genomic locus itself, which is beyond the technical scope of this work. Likewise, such systems can themselves perturb the localization of the target RNA. Another suggested approach, FISH, is also rife with technical caveats (see, for example: PMID 25555572). Fixation conditions can perturb localization, probe selection and validation can be technically cumbersome, and poorly vetted probes can lead to artefactual signal. Confounding this is that many of the novel RNAs we’ve identified are expressed at extremely low levels, making FISH quite challenging. Moreover, this analysis is complicated by a pervasive issue in the RNA localization field: the difficulty in comparing imaging and sequencing data (see, for example: PMID 25630241). Certainly, if we had identified abundant RNAs that are massively enriched in both the NLS and KDEL data sets, we would expect a FISH signal that would be highly localized near the lamina. But, it is difficult a priori to predict what the FISH signal of a lowly-expressed species that is modestly–though significantly–enriched at the lamina might appear, especially if this RNA is among the first of its class to be identified. We therefore believe that addressing this question in a robust way would require an extensive analysis that exceeds the scope of the current work. Finally, given the longstanding difficulties in identifying laminar RNAs, the mere generation of a speculative candidate list represents a significant achievement. We present this list as a compelling starting point for future analysis.

5) The analysis of RNAs potentially associated with the nuclear lamina seems problematic. Whereas 14% of the set of nuclear RNAs detected overlap with the ER-associated RNAs, only 6% of the lncRNAs overlap. Thus, the lncRNAs are under-represented in the population of nuclear RNAs that could be associated with the lamina. The caveat that RNAs identified as nuclear may not actually be nuclear further weakens this analysis.

While the reviewer’s comments are insightful, we offer the converse perspective on the data presented in Figure 5 . Namely, since the lamina is generally thought to be a transcriptionally repressive environment, we assume that laminarlocalized RNAs represent a minority of the nuclear-resident transcriptome. Nuclear RNAs are also quite diverse, comprising many classes that are involved in a multitude of functions. Hence, it might be misleading to try and identify laminar RNAs by looking for conspicuous enrichment of a single class of nuclear RNAs (e.g. lncRNAs) within the KDEL dataset. The majority of nuclear lncRNAs may be involved in broadly uncharacterized, non-laminar functions. In support of this, we know of relatively few lncRNAs that are conspicuously laminar. One such RNA, *NEAT1* (PMID: 25630241), is enriched in our KDEL RIP experiment.

Conversely, it may be more instructive to ask, of the KDEL-enriched RNA population, what proportions are also enriched from the nucleus? Looking at the data this way, we observe that, of the 2494 mRNAs enriched in the KDEL dataset, 385 (15%) are also enriched in the nucleus. For many of these RNAs we can further justify nuclear enrichment as a potential byproduct of regulated premRNA processing, as discussed above. However, of the 28 significantly enriched lncRNAs in the KDEL dataset, 11 (39%) are also enriched in the nucleus, a 2.5– fold higher proportion than that of their mRNA counterparts.

Finally, given the general agreement between the APEX-RIP-derived nuclearcytoplasmic data and those acquired by other methods (Figure 2, Author response image 1), we remain confident that the RNAs identified are nuclear.

**Author response image 1. respfig1:** Fractionation-Seq and APEX-RIP recover similar populations of intronic sequences from each compartment. A. Both methods quantify the nuclear enrichment and cytosolic de-enrichment of intronic reads (GENCODE hg19) to similar extents. We ascribe the greater number of intronic reads in the APEX pre-enrichment samples–relative to the Fractionation-Seq starting material–to our use of ribosome depletion, rather than poly(**A**) selection, in our starting samples (*methods*). B. Intronic reads exhibit similar subcellular partitioning by each method. Data shown are for all introns with starting RPKM≥1.0 in at least one experimental replicate. The data correlate with a Spearman’s r=0.88, *p*-value=0.

6) In the third paragraph of the Discussion, and at several other places of the manuscript, the authors make claims such as: "Compared to fractionation-based technologies, APEX-RIP offers superior accuracy". This seems like an over-interpretation based on the evidence presented, especially for the analyses of nucleo-cytoplasmic RNA distribution profiles. Firstly, the authors appear to draw this conclusion by comparing APEX-RIP and Frac-seq data from different cell lines (e.g. Figure 2). While it is likely that the nucleo-cytoplasmic localization of RNAs may exhibit a high degree of similarity between cell lines, comparison of results generated from the same cell lines would seem important to draw such conclusions.

We agree with the reviewer’s critique, and regret the oversight in our original analysis, which relied on nuclear-cytoplasmic RNA-seq data from the NHEK cell line. To address this concern, we have repeated this analysis using data derived from HEK 293T cells (Sultan et al., PMID: 25113896), the same line used in our experiments. While this is a much more reasonable basis for comparison, as the reviewer correctly anticipated, it has not substantially altered the experimental results. Our revised draft now focuses on these analyses.

Second, the detection of ER mRNAs within the nucleus in Frac-seq data, but not APEX2-NLS samples, could simply underline the ability of the Frac-seq approach to purify and detect pre-mRNAs. Do the authors detect such species within APEX2-NLS RNA-seq dataset? Is read coverage of intronic regions comparable to that observed with nuclear Frac-seq?

We agree with the reviewer and have addressed with further analysis. Specifically, by quantifying intronic reads in the Frac-Seq (Sultan et al.), NLS- and NES-APEX-RIP datasets, we observed that the nuclear-cytoplasmic distribution of pre-mRNAs was nearly indistinguishable between the two methods: relative to whole-cell RNA, intronic reads were enriched in the nuclear compartment, and de-enriched in the cytosol, to similar extents (Author response image 1). Furthermore, when we examined the partitioning of individual species between these compartments, we observed excellent agreement between the two methods (Spearman’s r=0.88; *p-*value ~0, Author response image 1). Hence, we conclude that, en masse, APEX-RIP and Frac-Seq are comparable approaches for isolating immature RNA species from the nucleus. This is also reflected in our revised analysis of the presence of ER-proximal RNAs in nuclear datasets, which shows general agreement between the methods (Figure 2).

7) The authors provide evidence that APEX-RIP captures RNAs with lower abundances than alternative approaches do, but it would be nice to have some sort of estimate of the limit of detection.

We have addressed this insightful point more robustly in the revised manuscript. Assigning a lower detection limit is often empirical, and absolutely measuring transcript abundances is technically challenging. However, we have tried to approach this problem with some degree of rigor in the revised Figure 3—figure supplement 2 , by calculating the expression range that encompasses 95% of the significantly enriched species in each ER dataset. For Frac-Seq and ribosome profiling, this range has a lower limit at 3.7 and 2.06 FPKM, respectively; for APEX-RIP this limit is 0.42 FPKM.

Furthermore, there is no discussion for any of the experiments about false negatives.

For each experiment, we characterize the coverage, or the recall of expected RNAs. The inverse of coverage is the false negative rate. Hence, if our coverage is 97.1% (for the “gold list” or “true positive list” of 71 mRNAs encoding established ER-resident proteins – Figure 4), then our false negative rate is 2.9% – the fraction of RNAs our method is expected to miss. We have now clarified this in the text and methods.

8) Regarding RNAseq library preparation and data analysis, it would be useful to include more detail in both the Materials and methods section and the main text to demonstrate that their sequencing datasets and the different reference datasets are indeed directly comparable. Specifically, in the Materials and methods section the authors should include the dataset IDs for the specific ENCODE datasets and explain whether they as well as the other mentioned datasets were generated using comparable library preparation and sequencing methods.

We agree and apologize for the oversight. We have revised the text to include the information in both the Materials and methods section and main text of the manuscript. We furthermore explicitly discuss the differences between this workflow and those used to generate the Frac-Seq and Ribosome Profiling data to which we compare our results, which no longer includes any ENCODE dataset. While the Frac-Seq libraries to which we compare our work were generated through methods comparable to our own, ribosome profiling is substantially different. These differences ultimately mean that ribosome profiling samples a different subset of the transcriptome (i.e.–ORFs within mRNAs). As discussed in the text, we believe that, by analyzing the data at the gene level (and not the transcript, ORF or µORF level), and by examining fold enrichments (which normalize the differences to which each RNA is sampled in a given workflow) rather than absolute abundances, we have accounted for these differences. Moreover, the fact that ribosome profiling and APEX-RIP sample different subsets of the transcriptome– namely, that APEX-RIP can analyze ncRNAs while ribosome profiling cannot–is part of the message of our work.

While the authors specifically mention lncRNAs, it is unclear how other, often highly abundant, non-protein coding RNAs were treated in the analysis. While this should be explained in detail in the Materials and methods, it would be useful to also include a brief description in the main text.

As per the reviewer’s suggestion, we have expanded the description of our library preparation and analysis workflows, and discuss in both the Materials and methods and main text the caveats to our approach. We enumerate the classes of RNA that may be opaque to our analysis, and why. Furthermore, the data presented in column C of Supplementary file 1, Supplementary file 2, Supplementary file 3 and Supplementary file 4 indicate the GENCODE-defined RNA class of each species analyzed.

9) For IF assays to confirm proper APEX/HRP fusion subcellular targeting, co-labeling with organelle specific markers should be shown.

Based on the reviewer’s suggestion, we have now included organelle-specific markers in the revised immunofluorescence Figure 1, Figure 3 and Figure1—figure supplement 2. DAPI serves to mark nuclei in Figure 2 and Figure 2—figure supplement 3 .